# Discovery of an agonistic Siglec-6 antibody that inhibits and reduces human mast cells

Julia Schanin[1], Wouter Korver[1], Emily C. Brock[1], John Leung[1], Zachary Benet[1], Thuy Luu[1], Katherine Chang[1], Alan Xu[1], Naomi De Freitas ![ORCID][1], Kenneth Luehrsen[1], Michael A. Brehm[2], Alan Wong[1] & Bradford A. Youngblood ![ORCID][1✉]

Mast cells (MC) are key drivers of allergic and inflammatory diseases. Sialic acid-binding immunoglobulin-like lectin (Siglec)-6 is an immunoregulatory receptor found on MCs. While it is recognized that engaging Siglecs with antibodies mediates inhibition across immune cells, the mechanisms that govern this agonism are not understood. Here we generated Siglec-6 mAb clones (AK01 to AK18) to better understand Siglec-6-mediated agonism. Siglec-6 mAbs displayed epitope-dependent receptor internalization and inhibitory activity. We identified a Siglec-6 mAb (AK04) that required Fc-mediated interaction for receptor internalization and induced inhibition and antibody-dependent cellular phagocytosis against MCs. AK04-mediated MC inhibition required Siglec-6 immunoreceptor tyrosine-based inhibitory motif (ITIM) and ITIM-like domains and was associated with receptor cluster formation containing inhibitory phosphatases. Treatment of humanized mice with AK04 inhibited systemic anaphylaxis with a single dose and reduced MCs with chronic dosing. Our findings suggest Siglec-6 activity is epitope dependent and highlight an agonistic Siglec-6 mAb as a potential therapeutic approach in allergic disease.

[1] Allakos, Inc., 825 Industrial Rd Suite, 500 San Carlos, CA, USA. [2] University of Massachusetts Medical School, Worcester, MA, USA.
✉email: byoungblood@allakos.com

Mast cells (MCs) are tissue-resident immune cells present in virtually all organs of the body, including those that interface with the external environment. MCs can play sentinel roles, coordinate immune responses and regulate both acute and chronic inflammation in many settings[1]. MCs are considered one of the most powerful immune cells based on their ability to respond to multiple stimuli and selectively release different types of mediators[2]. They are best known for their role in allergic responses, where they can be activated upon allergen-crosslinking of IgE bound to its high affinity receptor (FcεRI)[3–6]. Allergen activation of FcεRI triggers the release of pre-stored as well as newly synthesized inflammatory mediators that elicit allergic reactions ranging from rhinitis to anaphylaxis. In addition to FcεRI, MCs possess a myriad of activating cell surface receptors that include G-protein-coupled receptors (mas-related G protein-coupled receptor-X2 [MRGPRX2], chemokine and complement receptors), cytokine receptors (KIT, IL-4R), MyD88-dependent receptors (IL-33R, TLRs), and others[7].

Because of their location and unique biology, MCs are key drivers of many allergic diseases, including asthma, chronic urticaria, atopic dermatitis, eosinophilic gastrointestinal disease, and prurigo nodularis[4,8,9]. Importantly, MC mediators contribute to multiple inflammatory diseases, such as psoriasis, inflammatory bowel disease, osteoarthritis, chronic obstructive pulmonary disease, and others[10–12]. Based on the pathogenic nature of MCs, therapeutic approaches are urgently needed to selectively modulate MCs. While several clinically approved molecules target important MC activating receptors (FcεRI, IL-4R, IL-6R), and additional molecules are in development to deplete MCs, these approaches are not selective for MCs or have significant on-target toxicity[13]. One potential approach for targeting MCs is to engage inhibitory receptors that can silence MC activation, such as sialic acid-binding immunoglobulin-like lectins (Siglecs). Siglecs are a family of immune regulatory receptors primarily found on immune cells[14]. The majority of Siglecs contain immunoreceptor tyrosine-based inhibitory motifs (ITIMs) that function to counteract activating signals. Src-homology 2 (SH2) domain containing protein tyrosine phosphatases have been shown to be critical for inhibition of immune cells through Siglec family members and other ITIM containing receptors[15,16]. Interestingly, MCs express several inhibitory Siglecs, including Siglec-2 (CD22), Siglec-3 (CD33), Siglec-6, Siglec-7, and Siglec-8[17]. Previous studies have shown that engagement of CD33, Siglec-6, Siglec-7, and Siglec-8 with FcεRI induce inhibition of MC activation in vitro[17–21]. In addition, ligation of CD33 with ligand coated liposomes or Siglec-8 with a monoclonal antibody (mAb) reduces inflammation in vivo[22,23]. Indeed, lirentelimab (AK002), a humanized Siglec-8 mAb has shown beneficial activity in multiple clinical studies by depleting eosinophils and inhibiting MCs, suggesting Siglecs can modulate MC activation in humans[24–26].

Siglec-6 is an inhibitory receptor that is selectively expressed on human MCs and represents an attractive therapeutic target. Engagement of Siglec-6 with a mAb was recently shown to broadly inhibit MC activation in vitro[21], suggesting binding Siglec-6 with an agonist antibody leads to downstream inhibition. While it is well recognized that engaging Siglecs with agonistic antibodies or ligands mediates inhibition across immune cells, the mechanisms that govern this agonism are not well understood. We reasoned that a better understanding of Siglec-6 agonism as well as receptor biology would lead to the development of an optimal clinical antibody candidate to mediate potent and selective MC inhibition via targeting Siglec-6. Here, we report the generation of a panel of Siglec-6-specific mAbs binding to a range of extracellular domains to investigate the influence of epitope specificity on MC inhibition via Siglec-6 agonism. Siglec-6 mAbs displayed differential receptor internalization properties and inhibitory activity that were dependent on epitope. We identified a Siglec-6 mAb (clone AK04) that bound to a membrane-distal domain of Siglec-6, required Fc-mediated interaction for receptor internalization, and induced profound MC inhibition. AK04-mediated MC inhibition required functional Siglec-6 ITIM and ITIM-like domains and was associated with formation of immunoregulatory receptor clusters that contained SHP phosphatases. Importantly, treatment of humanized mice with AK04 fully inhibited systemic anaphylaxis with a single dose and significantly reduced human tissue MC numbers with chronic dosing.

## Results

**Siglec-6 mAbs display different binding characteristics.** Siglec-6 has been reported to be expressed on skin and esophageal tissue MCs, specific populations of trophoblasts, and memory B cells[21,27,28]. To confirm Siglec-6 expression, we profiled Siglec-6 surface expression on major immune cell populations in human peripheral blood as well as lung, skin, and gastrointestinal (GI) tissues by flow cytometry using a commercially available mAb (Supplemental Fig. 1a, b). Siglec-6 expression was consistently detected at high levels on MCs from all tissues evaluated (~6000 dMFI) (Supplemental Fig. 1c). In addition to MCs, low levels of Siglec-6 were found on unswitched (~250 dMFI) and switched (~450 dMFI) memory B cells (Supplemental Fig. 1c). Siglec-6 expression was not found on any other immune cells in blood or tissues.

To better understand the function of Siglec-6, we generated a panel of anti-human Siglec-6 mAbs using mouse hybridomas. The top 18 producing hybridoma clones (AK01-AK18) were expanded, subjected to variable region sequencing, and recombinantly produced on human and mouse IgG1 backbones. To evaluate binding specificity of the Siglec-6 mAbs to human Siglec-6, a cell-based human Siglec cross-reactivity assay was developed. Constructs encoding full-length DYK-tagged Siglecs (Siglec-3, 5, 6, 7, 8, 9, 10, 11, 14) were individually transfected and expressed on the surface of CHO cells. All transfected Siglecs were individually detected on the surface of CHO cells using an anti-DYK mouse antibody (Fig. 1a). Most of the Siglec-6 mAbs showed selective binding to human Siglec-6, whereas clones AK11, AK15, AK16, AK17, and AK18 displayed weak binding to all Siglecs tested (Fig. 1a). Next, we evaluated bivalent mAb avidity using biolayer interferometry for our panel of Siglec-6 mAbs. Most of the mAbs demonstrated high affinity to recombinant Siglec-6 extracellular domain (ECD) (Table 1). In contrast, clones AK15, AK16, AK17, and AK18 showed very weak to no binding, consistent with the cell-based screening assay. Interestingly, we found that the binding kinetics differed among specific Siglec-6 mAbs (Fig. 1b, Supplemental Fig. 2). The Siglec-6 mAb clones, AK01, AK03, and AK04 displayed reduced responses compared to other clones such as AK02 and AK05, suggesting they may have different binding properties.

Next, we set out to identify mAb binding sites within the ECD of Siglec-6. Wildtype (domain 1, 2, 3) and truncated (domain 1, 2 and domain 1) Siglec-6 ECD Fc-fusion proteins were generated to determine which domain the Siglec-6 mAbs each recognized (Fig. 1c). The panel of Siglec-6 mAbs bound across all three of the ECDs with most of the Siglec-6 mAbs binding to domain 1 (N-terminal ligand binding domain) or domain 3 (Ig-like domain) (Fig. 1d). To further define mAb binding sites, epitope binning, a technique used to cluster different mAbs by the epitope they recognize, was performed[29]. Epitope binning revealed that there were 5 distinct Siglec-6 mAb bins (A, B, C, D, E), with 3 of the bins consisting of mAbs that bind to domain 1 (Fig. 1e). Within domain 1, AK01, AK03, and AK04 were Bin A binders, AK05 and

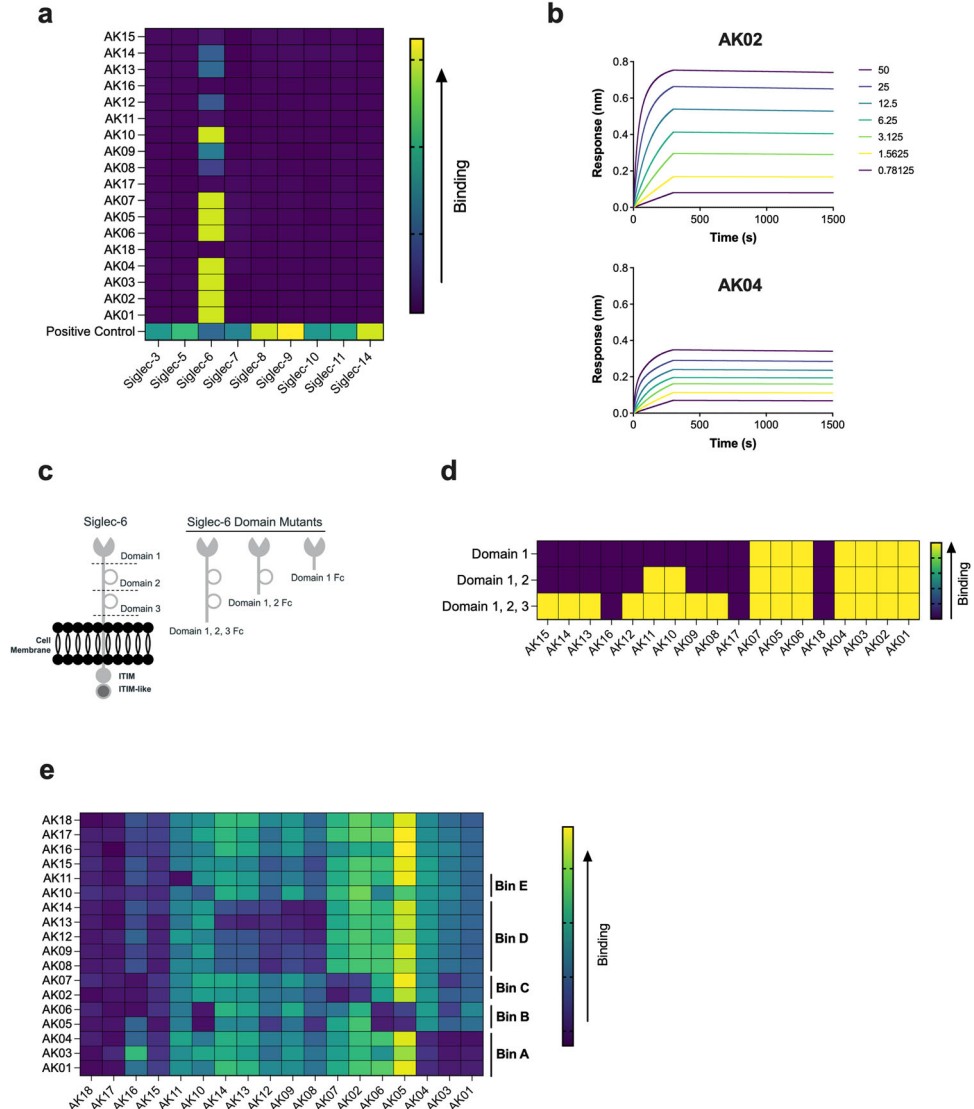

**Fig. 1 Siglec-6 mAbs display different binding characteristics. a** Heatmap representing binding of Siglec-6 mAbs to CHO cells transfected with individual Siglecs as determined by flow cytometry. **b** Assessment of Siglec-6 mAb avidity to Siglec-6 ECD by biolayer interferometry. Representative curves are shown for AK02 and AK04. **c** Schematic of Siglec-6 extracellular domains and domain mutants used for mapping. **d** Heatmap representing Siglec-6 mAb binding to Siglec-6 extracellular domains. Domain 1 binders bind to all constructs; domain 2 binders interact with domain 1, 2, 3, and domain 1, 2 constructs; and domain 3 binders only bind to domain 1, 2, 3, construct. **e** Heatmap displaying the 5 distinct epitope bins A, B, C, D, and E.

AK06 were Bin B binders, and AK02 and AK07 were Bin C binders. Bin A binders were the mAbs that had smaller binding response rates in Fig. 1b, further suggesting these clones may display differential activity compared to Bin B or C binding clones. Collectively, these data demonstrate our panel of Siglec-6 mAbs bind specifically to Siglec-6 and interact with the receptor in different locations with the N-terminal ligand binding domain being the most common and consisting of several unique epitopes.

**Siglec-6 mAbs show epitope-dependent receptor internalization properties.** To investigate if our panel of Siglec-6 mAbs mediated activity through Siglec-6, we first evaluated Siglec-6 mAb-induced receptor internalization by flow cytometry using peripheral blood-derived human MCs (hMCs). Siglec-6 internalization was detected using two fluorophore-conjugated, non-competing Siglec-6 mAbs (AK05 and AK02) depending on the treatment mAbs (Supplemental Fig. 3a, b). Strikingly, Siglec-6 mAb clones induced different levels of Siglec-6 internalization

assessed by titration studies (Fig. 2a). Clones within Bins B and C, including AK02 induced potent and complete receptor internalization, whereas Bin A clones, such as AK04 induced weak Siglec-6 internalization (Fig. 2a, b, Supplemental Video 1). Evaluation of the remaining Siglec-6 mAbs further highlighted an epitope-specific pattern of Siglec-6 receptor internalization with most of the clones in Bins B, C, D, and E showing strong internalization and those in Bin A displaying the weakest (Fig. 2b, Supplemental Fig. 3c). These data suggest that Siglec-6 receptor internalization is dependent on mAb binding location.

Fcγ receptor binding was recently shown to mediate programmed death-ligand 1 (PD-L1) receptor internalization for the PD-L1 mAb avelumab[30]. To evaluate if Fcγ receptor binding was needed for Siglec-6 receptor internalization for the Bin A binders, we cultured hMCs alone or in the presence of the Fcγ receptor expressing THP-1 monocytic cell line with AK04 hIgG1 or an isotype control. As expected, the THP-1 cells expressed multiple Fcγ receptors, including CD64, CD32, and CD16 (Supplemental Fig. 3d). Culturing hMCs with different ratios of THP-1 cells, but

**Table 1 Siglec-6 mAb characteristics.**

| Clone | Binding Domain | Epitope Bin | Affinity ($K_d$) |
|---|---|---|---|
| AK01 | 1 | A | $1.0 \times 10^{-12}$ |
| AK02 | 1 | C | $7.0 \times 10^{-12}$ |
| AK03 | 1 | A | $1.0 \times 10^{-12}$ |
| AK04 | 1 | A | $1.0 \times 10^{-12}$ |
| AK05 | 1 | B | $5.5 \times 10^{-11}$ |
| AK06 | 1 | B | $2.4 \times 10^{-11}$ |
| AK07 | 1 | C | $6.9 \times 10^{-11}$ |
| AK08 | 3 | D | $2.6 \times 10^{-10}$ |
| AK09 | 3 | D | $1.1 \times 10^{-10}$ |
| AK10 | 2 | E | $8.4 \times 10^{-11}$ |
| AK11 | 2 | E | $2.0 \times 10^{-12}$ |
| AK12 | 3 | D | $8.7 \times 10^{-11}$ |
| AK13 | 3 | D | $6.3 \times 10^{-11}$ |
| AK14 | 3 | D | $7.2 \times 10^{-11}$ |
| AK15 | 3 | ND | $1.0 \times 10^{-9}$ |
| AK16 | ND | ND | ND |
| AK17 | ND | ND | ND |
| AK18 | ND | ND | ND |

*ND* not detected.
$K_d$ dissociation constant.

not hMCs alone, induced receptor internalization in the presence of AK04 (Fig. 2c). To evaluate if AK04-mediated internalization in the presence of THP-1 cells was Fc-mediated, we generated F(ab')₂ fragments. In the presence of THP-1 cells, AK04 IgG, but not F(ab')₂ induced receptor internalization, demonstrating the Fc-region of AK04 is required for Siglec-6 internalization (Fig. 2d). In contrast, the Bin C binder AK02 induced Siglec-6 internalization independent of THP-1 cells or Fc-region (Fig. 2d). To understand the kinetics of AK04-mediated Siglec-6 internalization, hMCs were cultured in the presence of THP-1 cells and internalization was monitored over 24 h. Siglec-6 internalization occurred 1-h post-AK04 treatment and peaked around 4 h (Supplemental Fig. 3e). To confirm Fcγ receptors mediated AK04 internalization of Siglec-6, we co-cultured hMCs with Chinese hamster ovary (CHO) cells expressing the human high affinity IgG receptor, CD64 (CHO-CD64) (Supplemental Fig. 4a). AK04 induced dose-dependent Siglec-6 internalization in the presence of CHO-64 cells compared to an isotype control (Supplemental Fig. 4b), demonstrating Fcγ receptor interaction is required for AK04 internalization.

To ensure our Siglec-6 internalization findings using hMCs translated in vivo, we dosed humanized mice (NSG-SGM3) with human IgG1 AK02 or AK04 and measured Siglec-6 internalization across different MC populations. Humanized mice displayed Siglec-6 expression on MCs obtained from peritoneal lavage, skin, spleen, and lung tissues (Supplemental Fig. 4c). Consistent with our hMC findings, AK04 IgG but not F(ab')₂ induced Siglec-6 internalization of human tissue MCs (Fig. 2e). In contrast, both AK02 IgG and F(ab')₂ induced Siglec-6 internalization. These findings demonstrate that the Siglec-6 mAb AK04 requires Fc-interaction to cause receptor internalization but AK02 does not.

**Siglec-6 mediated MC inhibition and receptor clustering is epitope-dependent.** To investigate the inhibitory effects of our panel of Siglec-6 mAbs, hMCs were incubated with an agonistic anti-FcεRIα antibody (CRA-1). Titration of CRA-1 dose-dependently induced FcεRI-mediated MC activation as shown by increased CD63 expression (Supplemental Fig. 5a). To assess the inhibitory activity of the Siglec-6 mAbs, we used a single concentration of CRA-1 (250 ng mL⁻¹). Siglec-6 mAbs induced varying levels of hMC inhibition (Fig. 3a and Supplemental

Fig. 5b), in an inverse pattern that paralleled their ability to promote Siglec-6 receptor internalization. Notably, the antibodies with the strongest inhibitory activity were the Bin A binding clones that induced the least Siglec-6 internalization, such as AK04 (Fig. 3b).

To better understand Siglec-6 epitope-dependent agonism, we focused on AK02 and AK04 because these clones bind to domain 1 with similar affinities, but display different internalization properties, with AK04 requiring Fc-interaction. Using a single concentration of CRA-1, we further evaluated the inhibitory activity of AK02 and AK04 on IgE-mediated MC activation. AK04 significantly reduced degranulation of FcεRI-mediated MC activation compared to AK02 (Fig. 3c). Similar findings were seen through titration studies with both Siglec-6 mAb clones, confirming AK04 mediates more potent MC inhibition (Supplemental Fig. 5c). In addition to reduced degranulation, AK04 significantly decreased soluble mediator production of FcεRI-activated hMCs, including tryptase and IL-8 compared to AK02 (Fig. 3d). To determine if AK04 or AK02 treatment induced Siglec-6 clustering, we investigated the dynamics of Siglec-6 using live cell confocal imaging. Human MCs were imaged over time following treatment with AK04 and AK02 and cluster formation was monitored. While clusters were absent immediately after addition of anti-Siglec-6 mAbs, both clones rapidly induced clustering on the surface which peaked at 15 min (Fig. 3e). However, the size of clusters induced by AK04 were significantly larger than AK02, suggesting enhanced Siglec-6 agonism. Altogether, these data suggest that Siglec-6 mAb-mediated MC inhibition and receptor agonism are dependent on epitope.

**Siglec-6 ITIM and ITIM-like motifs are involved in AK04-mediated MC inhibition and phosphatase recruitment.** To better understand AK04-mediated MC inhibition and Siglec-6 receptor cluster formation, we transfected WT mouse bone marrow MCs (BMMCs) with human Siglec-6 expression constructs. Siglec-6 expressing BMMCs displayed similar expression across constructs and higher levels than hMCs (Supplemental Fig. 6a). Expression constructs were generated for mutant versions of Siglec-6, in which the tyrosine residues were replaced with phenylalanine residues, either separately (Y426F and Y446F) or combined to generate a double mutant (Y426F + Y446F) (Fig. 4a). BMMCs transfected with each of the WT or mutant Siglec-6 constructs were activated via anti-mouse FcεRI agonist antibody MAR-1 (isotype + anti-FcεRI). FcεRI activation was inhibited in the presence of AK04 for BMMCs transfected with WT or single-mutant Siglec-6 (Fig. 4b). However, BMMCs expressing the double ITIM mutant lost almost all Siglec-6-mediated inhibition. These data demonstrate that functional ITIM and ITIM-like motifs are required for Siglec-6 mediated inhibition in FcεRI-activated BMMCs.

Since the ITIM and ITIM-like motifs were required for Siglec-6 mediated inhibition in BMMCs, we assessed if Shp-1/2 physically interacted with Siglec-6 in BMMCs transfected with FLAG-tagged Siglec-6 (WT or mutant) and HA-tagged Shp-1 or -2. Wild-type Siglec-6 interacted with both Shp-1 and Shp-2, confirming the interaction between Siglec-6 and the inhibitory phosphatases (Fig. 4c, Panel B and C). Shp-1 interaction was also strongly detected with the single Siglec-6 ITIM mutants, but not with the double mutant. In contrast, the Shp-2 interaction was dependent on the proximal ITIM motif. To evaluate if the Shp-1 and -2/Siglec-6 interaction was dependent on Siglec-6 ITIM phosphorylation, we probed with a pan-phospho tyrosine antibody (Fig. 4c, Panel D). Indeed, WT but not the double mutant Siglec-6 was tyrosine phosphorylated. These data demonstrate that Siglec-6 interacts with both Shp-1 and Shp-2 and that this interaction is

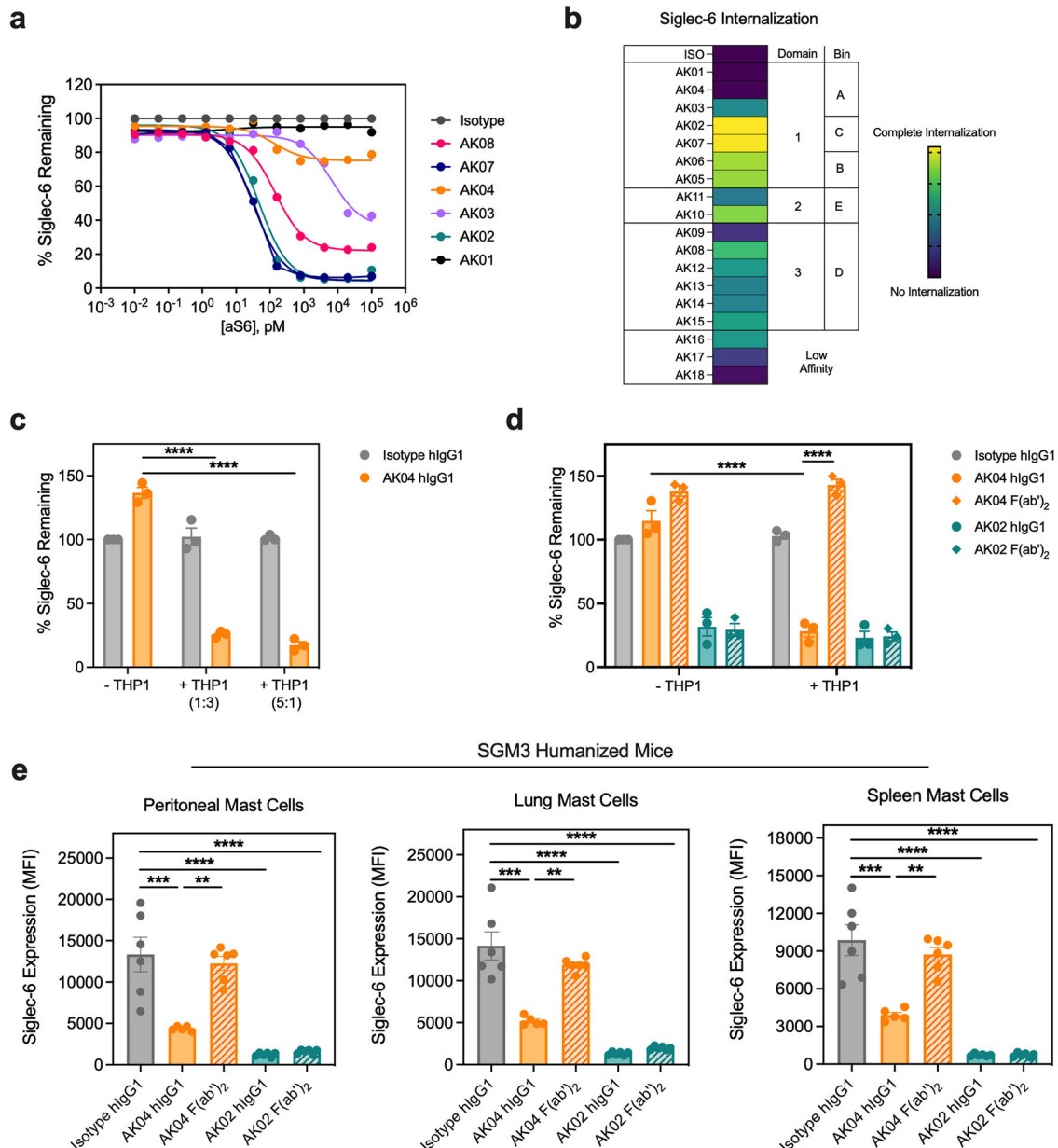

**Fig. 2 Siglec-6 mAbs show epitope-dependent receptor internalization properties. a** Siglec-6 internalization on hMCs after overnight incubation with the indicated Siglec-6 mAb clones as determined by flow cytometry using AK05 as the detection mAb. Data are representative of two experiments with two independent donors. **b** Heatmap of Siglec-6 receptor internalization levels induced by overnight incubation with Siglec-6 mAb clones (5 μg mL$^{-1}$) and their respective binding domains and bins. **c** Siglec-6 internalization on hMCs alone or in the presence of different ratios of THP-1 cells after incubation with 5 μg mL$^{-1}$ AK04 hIgG1 (orange) or isotype control (gray). **d** Siglec-6 internalization of hMCs alone or in the presence of THP-1 cells (1:3) after incubation with 5 μg mL$^{-1}$ of AK04 hIgG1 (orange), AK04 F(ab')2 (hashed orange), AK02 hIgG1 (green), AK02 F(ab')2 (hashed green), or isotype control (gray). **e** Siglec-6 expression on different MC populations from NSG-SGM3 humanized mice after intraperitoneal administration of 5 mg kg$^{-1}$ AK04 hIgG1 (orange), AK04 F(ab')2 (hashed orange), AK02 hIgG1 (green), AK02 F(ab')2 (hashed green), or isotype control (gray). Data are plotted as mean ± SD (three independent donors for Panel C and D; six mice/group for Panel E) and are representative of at least two experiments. **P < 0.01; ***P < 0.001; ****P < 0.0001 by two-way ANOVA with Šidák multiple-comparisons test.

dependent on Siglec-6 phosphorylation within the ITIM and ITIM-like motifs.

Because the ITIM and ITIM-like motifs were required for Siglec-6-mediated MC inhibition and phosphatase recruitment, we next investigated if AK04-induced receptor clusters contained these inhibitory molecules. BMMCs were transfected with WT or double ITIM mutant Siglec-6 expression plasmids, treated with a Siglec-6 mAb and subjected to confocal microscopy. In the untreated state, minimal co-localization of Shp-1 and Siglec-6 was seen in either WT or double ITIM mutant transfected BMMCs (Supplemental

Fig. 6b). Strikingly, Siglec-6 mAb-treatment resulted in Shp-1 co-localization in WT expressing Siglec-6 clusters, but not in the double ITIM mutant Siglec-6 clusters (Fig. 4d). These findings suggest that AK04-induced clusters contain inhibitory phosphatases required for downstream inhibition.

**Siglec-6 mAb treatment inhibits systemic anaphylaxis in humanized mice via MC inhibition.** To study Siglec-6 mAb-mediated human MC inhibition in vivo, we developed a passive

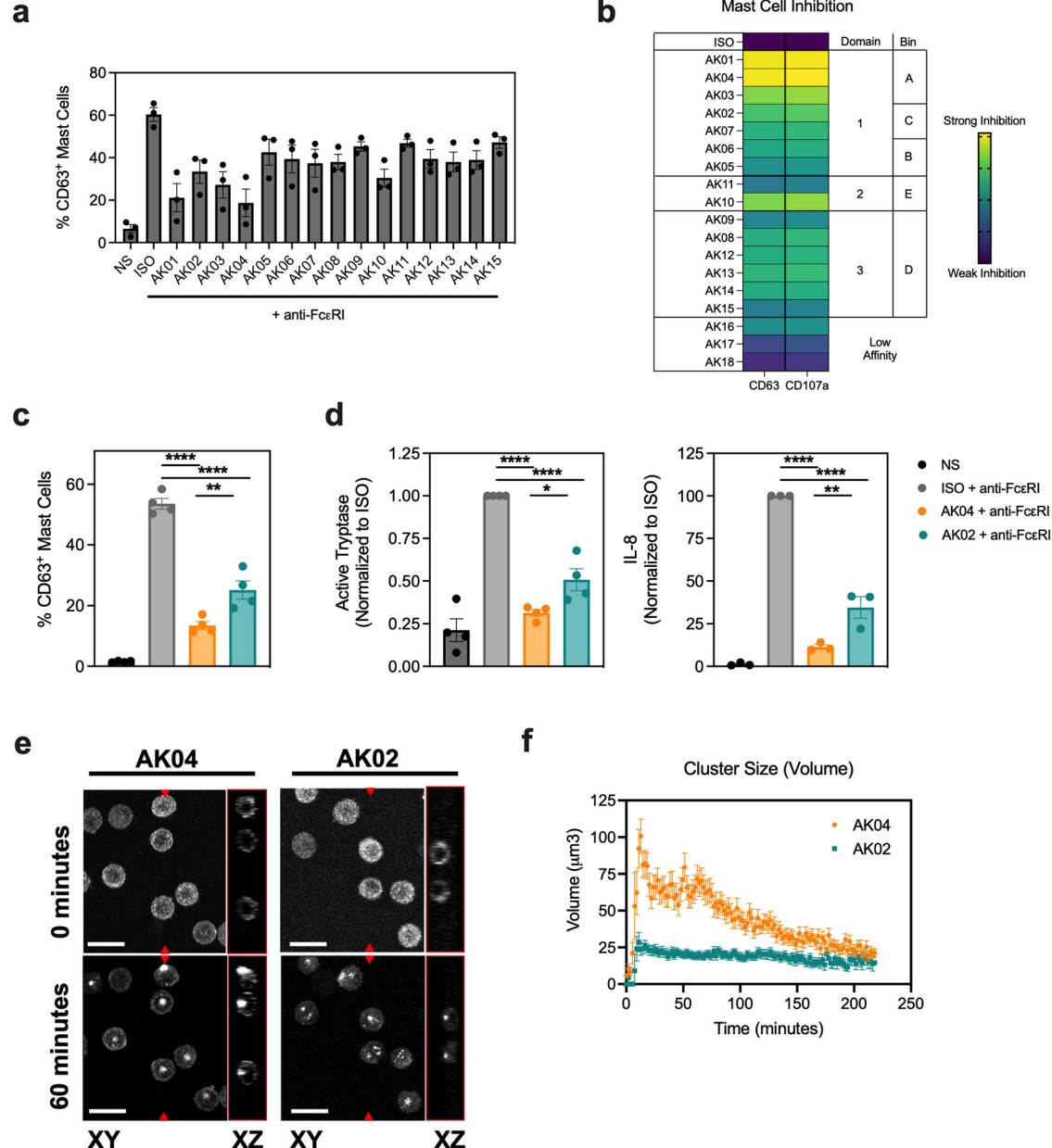

**Fig. 3 Siglec-6 mAb-mediated MC inhibition and receptor clustering is epitope dependent. a** Percentage of CD63+ hMCs non-stimulated or activated with anti-FcεRI antibody (CRA-1, 250 ng mL⁻¹) in the presence of Siglec-6 mAb clones (5 μg mL⁻¹) as determined by flow cytometry. **b** Heatmap of hMC inhibition induced by Siglec-6 mAb clones (5 μg mL⁻¹) and their respective binding domains and bins. **c** Percentage of CD63+ hMCs activated with anti-FcεRI antibody (CRA-1, 250 ng mL⁻¹) in the presence of 5 μg mL⁻¹ AK04 (orange), AK02 (green) or an isotype control (gray) compared to unstimulated hMCs (black). **d** Normalized supernatant levels of active tryptase and IL-8 from anti-FcεRI activated hMCs incubated with AK04 (orange), AK02 (green), or isotype control (gray). **e** Representative live confocal images of hMCs incubated with 5 μg mL⁻¹ AK04 or AK02 at 0 and 60 min. Scale bar = 15 μm. **f** Quantification of Siglec-6 cell surface clusters induced by AK04 (orange) and AK02 (green) over 250 min. Data are plotted as mean ± SD (three independent donors) and are representative of at least 2 experiments. *P < 0.05; **P < 0.01; ***P < 0.001; ****P < 0.0001 by two-way ANOVA with Šidák multiple-comparisons test.

systemic anaphylaxis model in NSG-SGM3 humanized mice, previously shown to be effective in BALB/c mice[31]. A single intravenous injection of the anti-human FcεRI antibody CRA-1 resulted in systemic anaphylaxis as characterized by a reduction in body temperature observed within 20 min after dosing (Fig. 5a). Notably, systemic anaphylaxis induced by CRA-1 administration was dose dependent. Injection of low dose CRA-1 lead to sustained lower body temperature up to termination at 60 min post challenge, whereas administration of high dose CRA-1 led to severe anaphylaxis resulting in death within 30 min (Fig. 5a).

To evaluate the in vivo inhibitory activity of a Siglec-6 mAb on human MC activation, AK04 was tested in a series of passive systemic anaphylaxis studies in NSG-SGM3 mice (Fig. 5b). Administration of a sublethal dose of CRA-1 resulted in a 4 °C drop in rectal temperature, which was completely prevented by dosing with a single injection of AK04 24 h prior to the CRA-1 challenge (Fig. 5c). The number of MCs in peritoneal lavage was not significantly affected by AK04 treatment in these studies (Fig. 5d), indicative of MC inhibition rather than depletion per se. Furthermore, AK04-treated mice showed significantly reduced

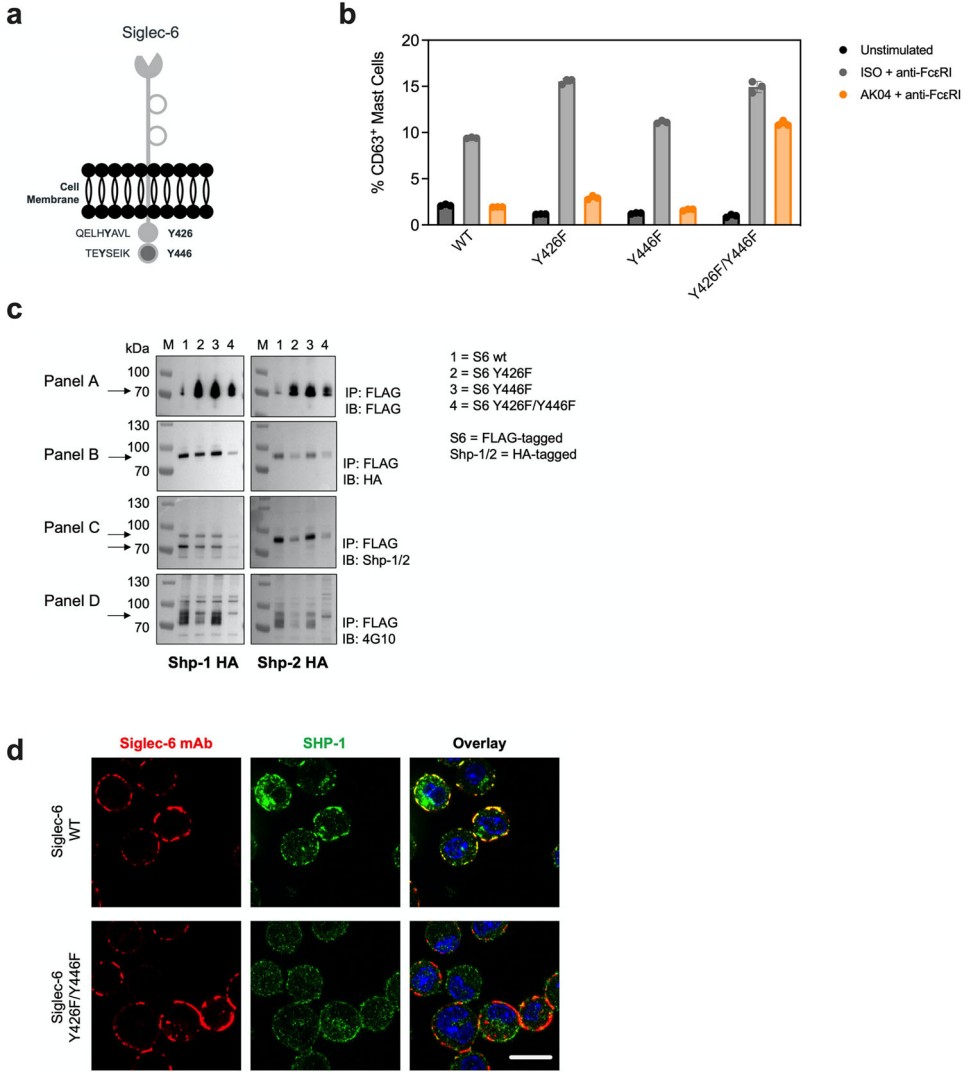

**Fig. 4 Siglec-6 ITIM and ITIM-like motifs are required for AK04-mediated MC inhibition and phosphatase recruitment. a** Schematic of the Siglec-6 receptor with its two tyrosine residues in the context of the proximal ITIM and distal ITIM-like motifs indicated. **b** Percent of CD63 positive BMMCs in unstimulated (isotype [ISO] + NS, S8 + NS), stimulated (ISO + anti-FcεRI) and inhibited (AK04 + anti-FcεRI) MC after transfection of the indicated WT or mutant Siglec-6 expression constructs Data are plotted as mean ± SEM ($n = 3$) and are representative of 2 experiments. **c** Western blot analysis of Siglec-6-Shp1/Shp2 interaction. Immunoblot (IB) of FLAG immunoprecipitations (IP) from BMMCs transfected with expression constructs for WT or mutant Siglec-6-FLAG and Shp-1-HA or Shp-2-HA. Cells were subjected to pervandate (PVD) treatment for two min and anti-FLAG IPs were analyzed for presence of Shp-1 (left column) or Shp-2 (right column), Siglec-6-FLAG (panel A), Shp-1/2-HA (panel B), Shp-1/2 (panel C) and p-Tyr (panel D). Experiments were performed 2-3 times for confirmation. **d** Confocal images of BMMCs transfected with WT or double mutant Siglec-6 constructs and treated with a labeled Siglec-6 mAb (red) for 45 min and stained with an anti-Shp-1 antibody (green). Scale bar = 10 μM.

levels of serum histamine and cytokines, including IL-4, IL-6 and CCL4 upon CRA-1 challenge compared to isotype control treated mice (Fig. 5e, f).

Next, we challenged a cohort of NSG-SGM3 mice with a higher CRA-1 dose to investigate the effect of AK04 on fatal anaphylaxis. Administration of a single dose of AK04 prevented death in the majority of animals, whereas most of the isotype mAb-treated control mice died within 20 min of CRA-1 injection (Fig. 5g). These results demonstrate that engagement of Siglec-6 with AK04 significantly prevents lethal anaphylaxis in humanized mice by inhibiting FcεRI-mediated MC activation.

**Siglec-6 mAb induces Fc-dependent ADCP against human MCs.** Because AK04 showed Fc-dependent Siglec-6 receptor internalization, we reasoned that this antibody property could promote antibody-dependent cellular phagocytosis (ADCP)

activity against MCs. To evaluate ADCP activity, hMCs were labeled with CellTrace and co-cultured with THP-1 cells in the presence of AK04 or an isotype control antibody (Fig. 6a). AK04 dose-dependently induced ADCP against hMCs in the presence of THP-1 cells (Fig. 6b). As expected, only AK04 hIgG but not F(ab')₂ triggered antibody-dependent phagocytosis, confirming the Fc-region of AK04 is required for Siglec-6 mAb-mediated ADCP activity of MCs (Fig. 6c, d). Since memory B cells had low, but detectable expression of Siglec-6, we next evaluated AK04-mediated ADCP activity of human primary B cells using THP-1 cells. As a positive control, we titrated rituximab, an anti-CD20 mAb with known ADCP activity against B cells[32]. Rituximab, but not AK04, induced ADCP of CD19+ B cells and reduced CD27+ memory B cell counts compared to an isotype control (Supplemental Fig. 7). These data demonstrate AK04 has potent and selective ADCP activity against MCs in vitro.

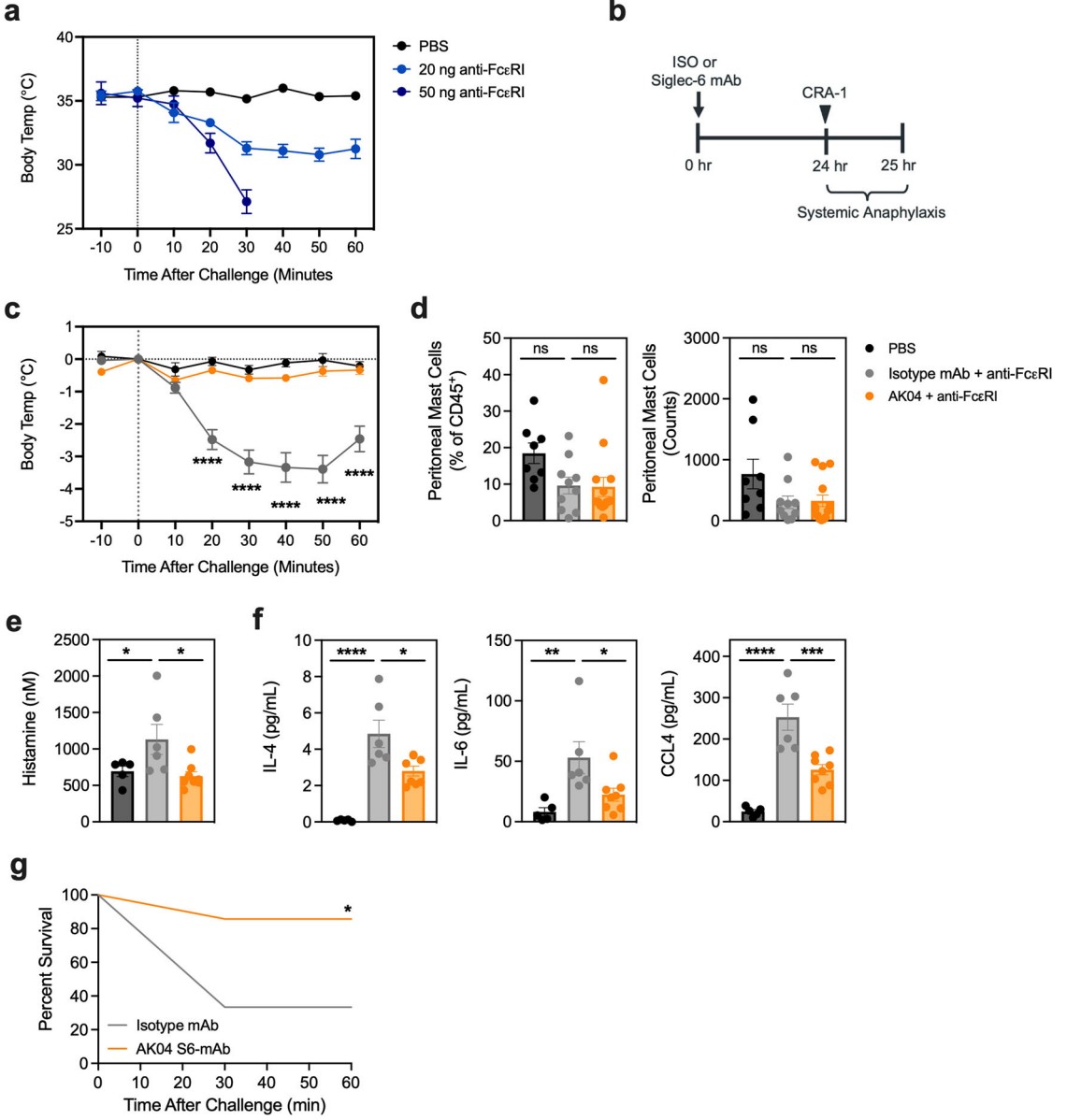

**Fig. 5 Siglec-6 treatment inhibits systemic anaphylaxis in humanized mice. a** Rectal body temperature of NSG-SGM3 humanized mice intravenously administered PBS (black), 20 ng anti-FcεRI, or 50 ng anti-FcεRI. **b** Schematic of sublethal passive systemic anaphylaxis experiment with AK04 or isotype control dosing arm. **c** Rectal body temperature of NSG-SGM3 mice intraperitoneally dosed with 5 mg kg⁻¹ AK04 (orange) or isotype control (gray) and 24 h later intravenously administered 20 ng anti-FcεRI or PBS (black). **d** Peritoneal MC percentages or counts as assessed by flow cytometry. **e** Histamine and (**f**) cytokine/chemokine levels in the serum of of NSG-SGM3 mice dosed with 5 mg kg⁻¹ AK04 (orange) or isotype control (gray) and 24 h later intravenously administered 20 ng anti-FcεRI or PBS (black). **g** Kaplan–Meier survival curve of NSG-SGM3 mice intravenously administered a fatal dose of anti-FcεRI (50 ng) after treatment with 5 mg kg⁻¹ AK04 (orange) or isotype control (gray) 24 h before. Data are plotted as mean ± SEM (Panel A: representative of three independent experiments with $n = 4$–5 mice/group; Panels C-F: pooled from 2 independent experiments with $n = 8$–15 mice/group; Panel G: $n = 6$–7 mice/group). *$P < 0.05$; **$P < 0.01$; ***$P < 0.001$; ****$P < 0.0001$ by two-way ANOVA with Šidák multiple-comparisons test. ns non-significant.

To evaluate if Siglec-6 mAb-treatment could reduce MC numbers in vivo, we repeatedly dosed NSG-SGM3 mice with AK04 hIgG1 or isotype control for 14 days followed by quantification of MCs in tissue using flow cytometry (Fig. 6e). AK04 treatment decreased human MC numbers in the peritoneal cavity, lung, and spleen by >50% compared to isotype control-treated mice (Fig. 6f). These results demonstrate that chronic dosing of AK04 reduces MCs in various tissue compartments in vivo.

## Discussion
MCs are powerful tissue-resident inflammatory cells equipped with a broad range of sensors that enable the recognition of myriad stimuli resulting in the release of pre-formed and newly synthesized inflammatory mediators. Because of their unique biology, MCs are considered key pathogenic cells in many allergic, pruritic, and inflammatory diseases. Despite their well-recognized role in disease pathogenesis, current MC-targeting approaches lack selectivity and/or broad inhibition. Due to their native inhibitory function and selective expression across immune cells, targeting the family of Siglec receptors, particularly Siglec-6, represents an intriguing strategy to selectively inhibit MCs via antibody agonism. Engagement of Siglecs with specific mAbs has been shown to inhibit B cells, monocytes, MCs, and eosinophils, highlighting mAbs that target Siglecs can function as

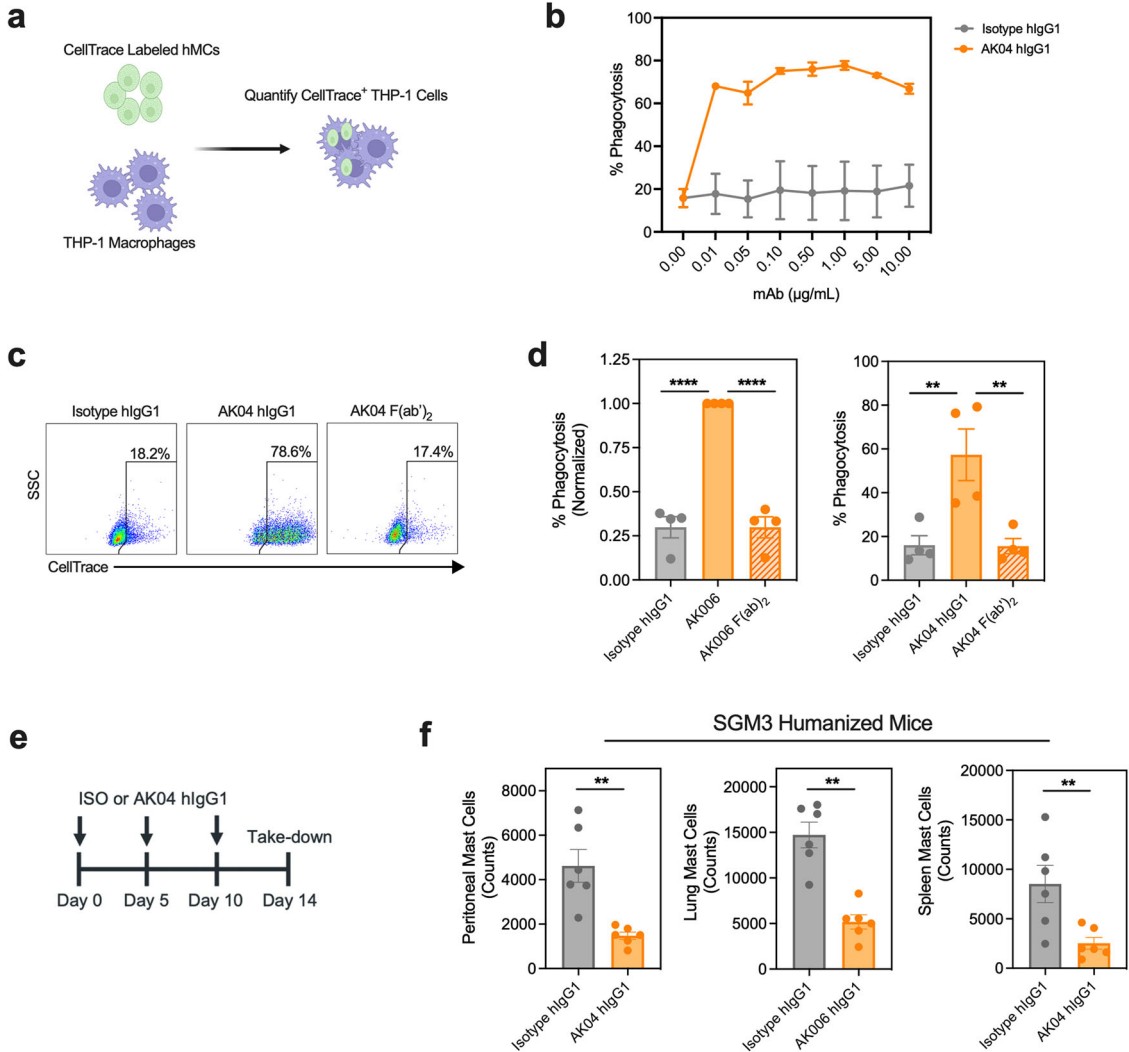

**Fig. 6 The Siglec-6 mAb AK04 reduces human MCs via antibody-dependent cellular phagocytosis. a** Siglec-6 mAb antibody-dependent cellular phagocytosis (ADCP) assay using hMCs and THP-1 cells. **b** Percentage of phagocytosis of hMCs after 4-h incubation with the indicated concentration of AK04 (orange) or isotype control (gray). **c** Representative dot plots of CellTrace positive THP-1 cells and (**d**) percentage of phagocytosis after 4-h incubation with AK04 hIgG1 (orange), AK04 F(ab')$_2$ (hashed orange), or isotype control (gray). **e** Schematic of 14-day dosing study in NSG-SGM3 humanized mice with AK04 or isotype control at 5 mg kg$^{-1}$. **f** MC counts from peritoneal cavity, lung, and spleen tissues at day 14 in NSG-SGM3 mice intraperitoneally dosed every 5 days with AK04 (orange) or isotype control (gray). Data are plotted as mean ± SD (Panel B: two independent donors; five independent donors for Panel C and D; 6 mice/group for Panel F) and are representative of at least 3 experiments. **P < 0.01; ***P < 0.001; ****P < 0.0001 by two-way ANOVA with Šidák multiple-comparisons test.

receptor agonists[33]. Nonetheless, the mechanisms that govern Siglec agonism are not well understood.

Our findings corroborate previous studies showing Siglec-6 is highly and selectively expressed on tissue MCs and to a much lower extent, memory B cells in non-malignant samples[21,27,28,34]. Despite detectable levels of Siglec-6 on B cells, Siglec-6 mAb treatment did not induce ADCP of memory B cells. These findings are consistent with the 'threshold' phenomenon, whereby high levels of surface antigen are required for optimal antibody-mediated effector mechanisms, such as ADCP and antibody-dependent cellular cytotoxicity (ADCC)[35].

Siglecs and many other cell surface receptors, such as PD-1 and PD-L1 are internalized upon mAb engagement[36,37]. Recently, Fcγ receptor binding was shown to be required for receptor internalization for the PD-L1 mAb avelumab[30]. Indeed, therapeutic strategies using antibody-drug conjugates have been developed to take advantage of this property for multiple Siglecs, including CD22, CD33, and Siglec-8[38–40]. Our findings suggest that mAb

binding location confers differential activity for Siglec-6, including receptor internalization and MC inhibition. Siglec-6 mAbs, like AK04 that bound to membrane-distal locations displayed reduced levels of receptor internalization compared to those binding closer to the membrane, such as AK02. However, in the presence of Fcγ receptor expressing cells, AK04 IgG but not F(ab')$_2$, induced Siglec-6 internalization, demonstrating that AK04 requires Fc-interaction to induce receptor internalization. We found that CD64, the high affinity Fcγ receptor was an important receptor for Siglec-6 internalization. Interaction with Fcγ receptors may enhance receptor density of Siglec-6 when engaged with AK04 thereby inducing internalization; however, this mechanism remains to be elucidated. While the addition of Fcγ receptor expressing cells was required for Fc-mediated receptor internalization for the hMCs, some tissue MCs express Fcγ receptors, which may be sufficient to induce receptor internalization on their own. To our knowledge, this is the first report of an anti-Siglec antibody that required Fc-mediated interaction for receptor internalization.

Our findings demonstrate that Siglec-6-mediated MC inhibition was also epitope dependent, consistent with other agonist antibodies against inhibitory receptors[41]. Interestingly, the Siglec-6 mAbs that showed the weakest receptor internalization induced the greatest MC inhibition. To investigate this further, we focused on two Siglec-6 mAb clones that both bound to domain 1 with similar affinity but displayed differential internalization activity. AK04 induced significantly greater MC inhibition and formed larger Siglec-6 receptor clusters than AK02. These observations are consistent with features of other agonistic antibodies targeting immunomodulatory receptors[41,42]. Antibody agonism of the TNF receptor (TNFR) family, such as CD40, induces receptor clustering that is dependent on the Fc portion of the antibody[43]. Fc-receptor binding has also been shown to be important for agonistic mAbs against CD28, CD32, and CTLA-4[44]. Receptor cluster formation and size are considered important features of agonistic mAbs as they are thought to represent immunoregulatory synapses that contain downstream signaling molecules[45]. Our findings using BMMCs align with these observations and show that AK04-induced receptor clusters contain inhibitory phosphatases that are dependent on functional Siglec-6 ITIMs. Mutation of the Siglec-6 ITIM and ITIM-like motifs also prevented AK04-induced MC inhibition, suggesting recruitment of phosphatases to Siglec-6 clusters is required for downstream inhibition. While additional studies are needed to elucidate the function of mAb-induced Siglec-6 clusters for MC inhibition, our findings suggest that mAbs targeting Siglecs can have differential agonistic activity that depend on many of the features described above.

Exploiting Fcγ receptors to mediate effector function is a common strategy for therapeutic mAbs. Engineering strategies have been focused on ADCC or ADCP which eliminate target cells through FcγRs on effector cells, such as NK cells or macrophages, respectively[41]. While both ADCC and ADCP are potential effector functions for a Siglec-6 hIgG1 mAb, we focused on ADCP since the cytotoxic NK cells that mediate ADCC are mainly found in blood and mature MCs are only found in tissue[46]. ADCP of human MCs using a Siglec-6 mAb represents an attractive approach to selectively reduce pathogenic MCs in disease settings. AK04 significantly reduced human MCs in the presence of macrophages. Similar findings were seen in chronic dosing studies with AK04 in humanized mice, suggesting the Fc-dependent property of AK04 yields additional effector function both in vitro and in vivo. It is interesting to note that AK04 mediates ADCP of MCs while also inducing Siglec-6 internalization. We hypothesize that the interaction between the Fc-region of AK04 and THP-1 cells is stabilized for a short period of time in which the THP-1 cells either phagocytose MCs or induce internalization. In support of this, AK04-induced internalization peaks around 4 h which should provide sufficient time for ADCP as this process has been reported to occur rapidly[47]. In addition, Siglec-6 is not completely internalized on the MC surface, providing additional opportunity for ADCP over time. However, additional studies are needed to better understand the kinetics of AK04-induced internalization and ADCP of MCs. While our findings support AK04 induces ADCP of mast cells via macrophages, we have not ruled out other mechanisms of macrophage activity, such as trogocytosis.

MC-targeting strategies have generally focused on neutralizing individual mediators or activating receptors on the cell surface. Strategies employing blockade of activating receptors expressed on MCs, including FcεRI, IL-4R, and thymic stromal lymphopoietin receptor (TSLPR) have shown promising clinical activity in many allergic diseases, in part by reducing MC activity[48]. Yet, the overall impact of MCs in a pathogenic setting is most likely multifaceted; that is, mediated by multiple activating mechanisms and diverse mediator production. Thus, targeting single pathways

on MCs may not be sufficient to broadly reduce MC activity[48]. To address these shortcomings, KIT targeting strategies have been developed to reduce MC numbers[49]. Indeed, administration of a single dose of the anti-KIT mAb leads to ablation of skin MCs and improvement of chronic urticarias[50]. However, KIT is expressed by cells other than MCs, including on hematopoietic stem cells, germ cells, and melanocytes[51–53]. Consistent with this expression profile, targeting KIT with a mAb or small molecule has been associated with neutropenia, graying of hair, changes in taste perception, and defects in spermatogenesis[50,54–56]. Our findings support that targeting Siglec-6 with an agonistic mAb to activate its native inhibitory function represents a needed approach to selectively target MCs. By generating a panel of well-characterized Siglec-6 mAbs, we confirmed that Siglec-6 is a powerful immunomodulatory receptor on MCs and identified an agonistic mAb that induces profound MC inhibition and reduces MC numbers through an Fc-dependent mechanism. These data highlight an agonistic Siglec-6 mAb as a potential therapeutic approach to selectively target and inhibit MCs in allergic and inflammatory diseases.

## Methods

**Human peripheral blood-derived mast cells**. Peripheral blood cells were isolated from residual cells in the leukocyte reduction chamber (TrimaAccel). Cells were eluted by gravity, and RBCs were lysed using 1X lysis buffer (BioLegend). CD34 + progenitor cells were isolated using the CD34 MicroBead UltraPure human KIT (Miltenyi Biotec) and cultured as previously described[57]. After 7 wk in culture, cells were maintained IMDM supplemented with 5% FBS, 55 μM 2-ME, 100 ng mL$^{-1}$ SCF, and 50 ng mL$^{-1}$ IL-6.

**Siglec-6 monoclonal antibody generation**. SJL/J mice were immunized with Siglec-6-ECD followed by fusion with HL-1 myeloma cells. Indivdual clones were selected based supernatant screening against human Siglec-6 via ELISA. The top 18 clones were selected and variable region sequencing was perfomed by PCR followed by recombinant production on mouse and human IgG1 backbones in Chinese hamster ovary (CHO) cells.

**Bivalent affinity determination and binning**. Binding affinities of Siglec-6 IgG for Siglec-6 ECD were measured by biolayer interferometry using a FortéBio Octet Red 96 instrument at 25 ℃ at 1000 rpm in 1X kinetics buffer (HEPES-buffered saline; GE Healthcare) in ultrapure water, with added stabilizer (FortéBio). IgGs were diluted from 12.5 to 0.78 nM in assay buffer in a 2-fold dilution series. Siglec-6 ECD-Fc biotinylated protein (Allakos, Inc, San Carlos, CA) was immobilized on streptavidin sensors at 100 nM in 1X kinetics buffer for 5 min until a sensor change of ~2.5 nm was achieved. The association phase was 5 min followed by a 20-min dissociation phase. An empty reference cell sensor was used as a blank control, and affinities analyzed using FortéBio analysis software with 1:1 global fit parameter. Epitope binning for the panel of Siglec-6 mAbs was done using a FortéBio Octet Red 96 instrument[29].

**Siglec cross-reactivity and domain mapping**. The CHO K1SV cell line (Lonza) was transfected with linearized plasmid DNA-encoding full-length human Siglec-3, 5, 6, 7, 8, 9,1 0, 11, and 14 using the Neon Transfection System. Cells were run through an Agilent Novocyte flow cytometer and analyzed with FlowJo software (Ashland, OR). CHO cells expressing full-length human Siglecs were plated at 100,000 cells per well, spun down at 400 g for 2 min, and washed once with FACS buffer (1% BSA in PBS). Cells were incubated with blocking solution (5% BSA in PBS) over ice for 30 min, followed by washing and incubation with 1 μg mL$^{-1}$ anti-Siglec-6 antibodies diluted in FACS buffer for 30 min on ice. Cells were then washed and incubated with 7.5 μg mL$^{-1}$ Ax647 conjugated donkey anti-mouse IgG (H + L) pAb diluted in FACS buffer with 7AAD viability dye.

For the domain mapping assay, maxisorp™ (Thermo Scientific) immunoassay microplates were coated with Goat anti-human Fc pAb (Thermo Scientific) at 1.0 μg mL$^{-1}$, incubated overnight, washed 4 times with 0.3 mL per well of PBS Tween-20 (PBST) (1× PBS [137 mM NaCl, 2.7 mM KCl, 10 mM Na$_2$HPO$_4$] plus 0.1% v/v Tween-20), and blocked with 0.3 mL of blocking buffer (2% BSA in PBST). Clarified CHO supernatants expressing the Siglec-6 domain fusion proteins were added to the wells and incubated for 1 h to capture the fusion proteins.

Bound Siglec-6 antibodies were detected with 0.5 μg mL$^{-1}$ donkey anti-mouse IgG (H + L)-HRP pAb, and fusion protein capture levels were detected using 0.5 μg mL$^{-1}$ donkey antihuman IgG (H + L)-HRP pAb for 1 h. After the final 1-h incubation, plates were washed with 1× PBST and 0.1 mL of 3,3′,5,5′-tetramethylbenzidine (Sigma) substrate was added to each well. After 2 min, the

reaction was terminated by adding 0.1 mL 1 M sulfuric acid. The converted substrate was detected in a spectrophotometer at 450 nm.

**Siglec-6 internalization assay.** Human MCs were plated in 96-well round bottom tissue culture plates at $5 \times 10^4$ per well and cultured overnight at 37 °C with either 5 μg mL$^{-1}$ or specified concentration of a Siglec-6 mAb or isotype control. The next day, residual Siglec-6 on the surface was detected by flow cytometry using a non-blocking Siglec-6 mAb conjugated to Alexa Fluor® 647 at 5 μg mL$^{-1}$. AK05 was used as the detection antibody for all Siglec-6 mAb clones except AK05 and AK06. To assess internalization of AK05 and AK06, the Siglec-6 mAb AK02 was used as the detection antibody. MFI data were normalized to isotype control and plotted as percentage of Siglec-6 remaining on the surface of MCs compared with isotype control antibody at the same concentration. To identify Siglec-6 mAbs that could be used to detect internalization, hMCs were incubated at 4 °C with 5 μg mL$^{-1}$ of the indicated clone for 30 min followed by detection of Siglec-6$^+$ cells by either AK02 or AK05 by flow cytometry. In co-culture experiments with THP-1 cells (ATCC) or CHO cells stably expressing human CD64 (GenScript), human MCs were treated the same way with antibodies but cultured overnight in the presence of the indicated number of THP-1 or CHO-64 cells.

**Human mast cell activation assay.** Human MCs were plated in 96-well round bottom tissue culture plates at $5 \times 10^4$ per well and centrifuged for 2 min at 400 g prior to resuspending with anti-FcεRI (clone CRA-1, Miltenyi Biotec) at 250 ng mL$^{-1}$ plus isotype-control mouse antibody (MOPC21, Allakos Inc) or Siglec-6 mAbs (Allakos Inc) at 5 μg mL$^{-1}$ or indicated concentration at 4 °C for 2 min. Cells were washed in PBS and then incubated in PBS with 10 μg mL$^{-1}$ secondary antibody (Thermo). After an additional PBS wash, cells were resuspended and incubated for 20 min at 37 °C for flow analysis or for 6 h for analysis cytokine levels in the supernatant. The percent of CD63 (Biolegend, 353004) and CD107a (Biolegend, 328608) expressing cells was determined by flow cytometry on a Novocyte Quanteon (Agilent). For cytokine quantification, 25 μl of supernatant was analyzed using Meso Scale Discovery's U-plex KIT (customized for the quantification of the indicated cytokines). Active tryptase activity was determined using the mast cell degranulation assay kit (Millipore Sigma).

**Passive systemic anaphylaxis model in humanized mice.** Female NOD.Cg-$Prkdc^{scid}$ $Il2rg^{tm1Wjl}$ Tg(CMV-$IL3$, $CSF2$, $KITLG$)1Eav mice (NSG-SGM3) were purchased from Jackson Laboratory engrafted with $5 \times 10^5$ human cord blood CD34$^+$ human hematopoietic stem cells and were used for in vivo studies as previously described[58]. Systemic anaphylaxis was induced by intravenously administering 20–50 ng of an agonistic anti-FcεRI antibody (CRA-1, Miltenyi Biotec)[58,59]. Anaphylaxis was defined as a significant decrease in core body temperature as measured by rectal probe immediately after challenge, every 10 min for 60 min. Mice were dosed i.p. at 5 mg kg$^{-1}$ with either an isotype control mAb (MOPC21, Allakos Inc, San Carlos, CA) or Siglec-6 mAb (AK04, Allakos Inc, San Carlos, CA) 24 h before CRA-1 challenge. After completion of body temperature measurements, peritoneal lavage and serum were harvested for MC quantification by flow cytometry and mediator analysis, respectively. Histamine levels were determined using an enzyme immunoassay kit (Beckman Coulter). For cytokine quantification, 25 μl of supernatant was analyzed using Meso Scale Discovery's U-plex KIT (customized for the quantification of the indicated cytokines). Muri-Genics Biosciences (Vallejo, CA) performed the in-life portion of these studies. All mice were housed in a specific pathogen-free environment.

**Peripheral blood and tissue processing.** Fresh human whole blood in purple top tubes was obtained from healthy donors at the Stanford Blood Bank. Blood was processed through two rounds of red blood cell (RBC) lysis in 1x RBC lysis buffer (eBioscience 00-4300-54), according to manufacturer instructions. The cell pellet was washed twice with 50 mL phosphate-buffered saline (1x PBS), centrifuged, suspended in RPMI-1640 containing 10% fetal bovine serum, and passed through a 40 μm nylon filter. Cell viability was examined using flow cytometry. Only single-cell suspensions that had at least 80% viability were used in subsequent experiments. Fresh human lung, skin, and GI tissue was procured and provided by the NCI Cooperative Human Tissue Network (CHTN) from subjects with no previous history of chronic disease. Human and humanized mouse tissues were enzymatically and mechanically dissociated using the gentleMACs™ Dissociator (Miltenyi Biotec), according to manufacturer's protocol. Peripheral blood leukocytes isolated from healthy blood and single cell tissue suspensions were seeded in U-shaped-bottom 96-well plate (Falcon Cat: 353077) at $3 \times 10^5$ cells/well.

**Flow cytometry.** Approximately $1–5 \times 10^6$ cells were preincubated with mouse CD16/32 antibody (Fc Block, BD Biosciences) for 10 min at 4 °C to block non-specific binding. Cells were then incubated at 4 °C for 10 min with staining antibody panels at 1 μg mL$^{-1}$ unless stated differently, washed, and fixed in 2% paraformaldehyde. Data acquisition was performed using a NovoCyte flow cytometer (Acea Biosciences) and FlowJo (San Diego, CA) was used for data analysis. The following antibodies were used for gating human immune cells and were purchased from eBioscience, Biolegend, R&D Systems, or Miltenyi Biosciences

(catalog number or clone indicated in brackets): CD11b (M1/70), HLA-DR (340549), CD45 (368528), CD117 (130091733), CD14 (301834), CD15 (555401), CD16 (557744), CD3 (OKT3), CD11c (301606), FcεR1 (CRA-1), CD19 (302206), CD20 (302330), CD38 (303532), CD27 (356410), IgD (348210), and Siglec-6 (MAB2859).

**FcεRI-mediated BMMC activation and anti-S6 treatment.** BMMCs expressing Siglec-6 were plated in 96-well round bottom tissue culture plates at $5 \times 10^4$ per well and centrifuged for 2 min at 400 g prior to resuspending with biotinylated anti-FcεRI (clone MAR-1, Biolegend) at 250 ng mL$^{-1}$ plus biotinylated isotype-control mouse antibody (MOPC21, mouse IgG1, Allakos) or biotinylated Siglec-6 mAb (AK04 mouse IgG1, Allakos) at 5 μg mL$^{-1}$ at 4 °C for 2 min. Cells were washed in PBS and then incubated in PBS with 10 μg mL$^{-1}$ neutravidin (Thermo) for 2 min. After an additional PBS wash, cells were resuspended in 200 μl 37 °C complete medium and incubated for 20 min at 37 °C for flow analysis or for 1 or 6 h for analysis of histamine or cytokine levels in the supernatant. For flow cytometry, cells were resuspended in 100 μl cold FACS buffer (PBS/1%BSA) containing 100 ng anti-CD63-PE/Cy7 antibody (clone NVG-2, Biolegend), 3 μl 7-AAD (Becton Dickinson) as viability marker and 0.2 μl mouse Fc block (BD). The percent of CD63 expressing cells was determined by flow cytometry on a Novocyte Quanteon (Agilent).

**Antibody-dependent cellular phagocytosis assay.** Human MCs or isolated human B cells from healthy human donors were labeled with 1 μM CellTrace Violet (Invitrogen C34557) according to manufacturer instructions and plated in 96-well round bottom tissue culture plates at $3 \times 10^4$ per well. Activated THP-1 cells that had been pretreated overnight with 50 ng mL$^{-1}$ of IFNγ were added to the mast cell culture at the indicated amount (typically $1 \times 10^4$). Cells were cultured at 37 °C with either 0.1 μg mL$^{-1}$ or specified concentrations of a Siglec-6 mAb, isotype control, or rituximab (Bioxcell). After 4 h, cells were stained with mast cell markers FcεRI and CD117 and residual Siglec-6 on the surface was detected by flow cytometry using a non-blocking Siglec-6 mAb conjugated to Alexa Fluor® 647. Percent Phagocytosis was measured by the percentage of THP1 cells that were CellTrace +. The percent CellTrace+ was normalized to Siglec-6 mAb-treatment and compared with isotype control antibody at the same concentration. Human B cells were isolated using negative selection beads (Miltenyi Biotec).

**BMMC transfection, immunoprecipitation and western blotting.** BMMCs ($2 \times 10^6$), generated from C57BL/6 mice as described[19], were transfected with 10 μg plasmid expressing WT or mutant full-length Siglec-6 fused with an N-terminal FLAG tag. In addition to the WT sequence, plasmids containing ITIM and ITIM-like mutants of Siglec-6 were generated by changing the tyrosine residue at position 426 of the proximal motif (WT = QELHYAVL) to phenylalanine (Y426F = QELHFAVL) and the tyrosine at position 446 of the distal motif (WT = TEYSEIK) to phenylalanine (Y446F = TEFSEIK). The double mutant contains both mutations (Y426F + Y446F).

**Transfection.** For all transfections, plasmid DNA (10 μg) was added to $2 \times 10^6$ cells and transfected using the 4D-Nucleofector (Lonza) with P3 nucleofector solution and Supplement 1 and program DS-130. Transfection efficiency of BMMC was consistently between 80–95% under these conditions. Cells were incubated at 37 °C in complete medium for 6 h for lysis for IP/WB or 16 h for functional assays (FcεRI cross-linking). A fresh 30 mM stock solution of pervanadate was prepared by adding 150 μl of 200 mM sodium orthovanadate (FivePhoton/Fisher) to 844 μl PBS, plus 6.1 μl 30% $H_2O_2$ (Sigma) and incubated for 20 min at room temperature prior to addition to cells at the indicated final concentrations to inhibit phosphatases.

**Immunoprecipitation and Western Blotting.** Cells were lysed in Pierce IP lysis buffer (Thermo) with HALT protease and phosphatase inhibitors (Thermo). Lysates were used directly for Western Blotting or IP by adding 25 μl Pierce anti-FLAG or anti-HA magnetic agarose beads (Thermo) for 30 min at room temperature while rotating. After 4 washes with TBS-T, proteins were eluted by incubation at 70 °C for 10 min in 1xLB with reducing agent added (Thermo), separated by SDS-PAGE and subjected to Western Blot analysis. 5% milk (Bio-Rad) in TBS-0.1%Tween-20 was used for blocking. Anti-FLAG antibody was from Sigma; anti-HA, anti-Shp-1, anti-Shp-2 (clone D50F2) from Cell Signaling Technologies; 4G10 from Millipore/Sigma. Anti-FLAG and anti-4G10 Abs for Western blotting were directly conjugated to HRP, other primary antibodies were detected using HRP conjugated mouse anti-rabbit IgG, light chain specific (Jackson ImmunoResearch 211-032-171) at 1:2,000 dilution in TBS-T/5% milk.

**Confocal microscopy.** Human MCs in phenol-red free RPMI were plated onto iBidi 96-well plates at a density of $10^5$ per well. Cells were then treated with a final concentration of 100 ng mL$^{-1}$ of AK02 or AK04 conjugated to Alexa-647. Images were acquired every 110 s using a Leica Stellaris 5 Confocal Microscope, an Oko Labs environmental chamber (37 C and 5% CO2), and a 40x objective. Two regions per well were acquired per experiment. Z stacks were acquired at an interval of

2.5 μm. Images were analyzed for clusters using ImageJ/FIJI. BMMCs expressing Siglec-6 ($10^5$) were transferred to iBidi plates in PBS and incubated at 37 °C for 30 min, prior to treatment with AF647-conjugated anti-S6 mAb at 10 μg mL$^{-1}$ for 0 min (unstimulated) or 45 min. After spinning at 350 g for 2 min and two washes with PBS, cells were fixed in 4% PFA in PS for 15 min at room temperature, followed by two more PBS washes and permeabilization in 0.2% Triton X-100 (Sigma) in PBS for 15 min at room temperature. After two washes, blocking was 1 h with 3% BSA in PBS and staining with anti-Shp1 Ab (clone 3H20L13, Thermo) for 16 h at 4 °C in 0.2% Triton X-100/1% BSA/PBS. After two more washes in PBS, goat anti-rabbit-AF488 (Jackson Immunolabs) was used for staining at 1:1000 in 0.2% Triton X-100/1% BSA/PBS for 1 h. Washed cells were overlaid with PBS containing 0.05 mg mL$^{-1}$ DAPI (Thermo) before imaging with a 63x objective.

**Statistics and reproducibility**. To determine statistical significance, nonparametric Mann Whitney U test, unpaired 2-tailed Student's t test, 2-tailed t test with Sidak's post-test, or one-way ANOVA with Tukey's post-test for multiple comparisons were performed using GraphPad Prism (GraphPad Software). A P value of 0.05 or less was considered significant. Human blood cells were collected from 3–5 independent donors. Sample sizes for experiments were 6–15 mice per group, and data are representative of at least 2 experiments.

**Study approval**. The animal studies were approved by the MuriGenics Biosciences Institutional Animal Care and Use Committee (IACUC) Animal Use and Care Committee and complied with the Guidelines for Care and Use of Laboratory Animals issued by the USA National Institute of Health.

**Reporting summary**. Further information on research design is available in the Nature Portfolio Reporting Summary linked to this article.

## Data availability

Source data underlying main figures are provided in Supplementary Data 1. Uncropped and unedited images of the blots that appear in the main article are provided in Supplemental Fig. 8. Allakos materials described in this manuscript may be available to qualified academic researchers upon request. In certain circumstances in which we are unable to provide a particular proprietary reagent, an alternative molecule may be provided that behaves in a similar manner.

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

## Acknowledgements

The study was funded by Allakos, Inc. We thank Drs. Bruce Bochner and Robert Schleimer for critical reading of the manuscript. Jocelyn Hybiske, PhD, an independent, professional medical writer, provided editing services that were funded by Allakos Inc.

## Author contributions

J.S., W.K., J.L., M.A.B., and B.A.Y. designed experiments; J.S., W.K., J.L., E.C.B., Z.B., T.L., K.C., A.X., N.D.F., and A.W. conducted experiments and acquired/analyzed data; K.L. provided molecular biology services; J.S., W.K., and B.A.Y. wrote the manuscript.

## Competing interests

These authors declare the following competing interests: JS, WK, ECB, JL, ZB, TL, KC, AX, NdF, KL, AW, and BAY are or were at the time the work was conducted employees of and own stock and/or stock options from Allakos, Inc. The remaining authors declare no competing interests.
