## [Peer Review File · Communications Biology]

Reviewers' comments:

Reviewer #1 (Remarks to the Author):

This manuscript describes development of 14 monoclonal antibodies specific for Siglec-6 (Sig6), an inhibitory receptor that is preferentially expressed on mast cells. The antibodies are compared for their affinity for Sig6, their specificity for binding to one of the three Ig domains of Sig6, and for overlap in their binding footprints when binding to the epitope they recognize (epitope binning). The work nicely documents the affinities of the antibodies and shows that they differ in their ability to induce internalization of Sig6. One of them, AK04, is investigated in detail for its properties for the internalization of Sig6 and modulation of mast cell signaling. While the work on characterization of these antibodies is of interest, aspects of the characterization of AK04 are not clear.

Specific points:

1. In Figure 2A several antibodies are shown to induce complete endocytosis/internalization of Sig6 while others show effectively none (e.g. AK01 and AK04). Residual Sig6 remaining on the surface was detected using a commercially available 'non-blocking' anti-Sig6 antibody. The conclusions rely on the commercial antibody (AF647-labeled) to have no overlap in binding to Sig6 with the other antibodies. A simple control would be to do the experiment at 4° C where no internalization would be possible (e.g. 1 hr at 4°). If there is no competition, the AF647-anti-Sig6 binding should be the same for all antibodies. Otherwise, the data in Fig. 2A could simply be a result of some antibodies preventing binding of the AF647-anti-Sig6.
2. The rationale for the internalization experiment with THP1 cells in Fig. 2C is not clear. AK04 was just demonstrated not to be an internalizing antibody in Fig. 2A. Now, it is said to internalize dependent on Fcγ receptors. Presumably this is meant to be Fcγ receptors on THP1 cells. Not clear how receptors on another cell can mediate internalization UNLESS it is meant that the THP1 cells phagocytose the mast cells. This is actually proposed in Fig. 6A. There is no clear explanation concerning how Fcγ receptors on THP1 cells can mediate AK04 internalization/endocytosis of Sig6.
3. In Fig. 4, how does the expression of Sig6 in BMMCs compare to expression of human mast cells.
4. Does AK04 reduction of anaphylaxis in Fig. 5 relate to the induction of phagocytosis by THP1 cells in Fig. 6? The mast cell counts in Fig. 5 are similar for AK04 and IsoType antibody, so it suggests not.
5. (minor) No vendor is listed for CRA-1.

Reviewer #2 (Remarks to the Author):

In their manuscript, Schanin et al. investigated MC inhibition by engagement of MC-specific Siglec-6 by antibodies. The Authors generated a panel of anti-Siglec-6 mAbs which they tested on cells transfected with Siglec-6 by analyzing their binding properties. Next, they evaluated the functionality of these antibodies on human MCs by investigating receptor internalization, in presence or absence of THP-1 cells expressing Fcγ-bearing receptors, and MC activation inhibition by engagement with 2 antibodies from the panel they created. Moreover, the Authors tested the effect of the chosen antibodies in mouse models of passive cutaneous anaphylaxis. Overall, the results presented are interesting and the manuscript is well written. However, before being published, there are several points to be addressed:

- 1) The first results presented, regarding Siglec-6 expression on MCs from different cells/tissues, seem to be somehow redundant, as partially shown already in Robida et al., 2022. In addition, these results do not appear to be in line with the concept of the manuscript, which shows the characterization of

these new anti-Siglec-6 antibodies. The first paragraph of the results should be included in the subsequent one, as proof of concept and reconfirmation that Siglec-6 is expressed specifically on MCs.

- 2) In the MC activation assay, the Authors tested only one concentration of the anti-Siglec-6 antibody. It is possible that the lack of significant inhibition seen with some of the antibodies might be due to not sufficient concentrations of the antibodies themselves. The Authors should discuss this point.
- 3) It is not clear why the Authors chose to employ the AK02 antibody, considering that AK01 showed better inhibition of MC activation. This point has to be discussed.
- 4) Did the Authors check the release of soluble mediators from MCs? This would be important to understand whether engagement of Siglec-6 by mAbs influences MC degranulation.
- 5) How do the Authors explain MC ADCP, when co-culture of MCs and THP-1 cells resulted in Siglec-6 internalization? How is phagocytosis of MCs triggered, if the antibody is internalized with the receptor? The Authors should discuss this point.
- 6) Did the Authors evaluate MC repopulation in the tissues after depletion with the antibody?
- 7) The Authors detected Siglec-6 only on MCs but not on other immune cells in the tissues, and on trophoblasts, which was previously reported as a human-specific occurrence (Brinkman-Van der Linden et al., 2007). The Authors should discuss more the fact that Siglec-6 expression in humans is so specific.

Reviewer #3 (Remarks to the Author):

Siglec-6 is an immunomodulatory cell surface receptor found on mast cells and mast cells are relevant in many pathological conditions, especially with respect to allergies. The authors set out to develop an antibody to target and internalize Siglec-6. The authors developed a panel of antibodies that targeted different epitopes of Siglec-6 and determine that the epitope of the antibody conveyed specific properties with respect to internalization to that antibody. The authors also evaluated the initial therapeutic potential of their antibodies and found that their antibody decreased allergic response in a mouse allergy model. Interestingly, they found that their antibodies function was dependant on the Fc receptor and the "ITIMs" of Siglec-6. The authors claim to be the first to demonstrate the requirement of the Fc receptor for internalization driven by an antibody for a Siglec and that their antibody may have therapeutic potential for the reduction of anaphylaxis. While this is an interesting study, I have a several major and minor concerns that need to be addressed:

Major:

1. The authors have missed that Siglec-6 is also expressed on memory B cells. Depleting memory B cells would completely erase immunological memory and be a major limitation of anti-Sig6 dependent depletion of mast cells. At minimum, 'off-target' effects on memory B cells need to be assessed if an anti-Sig6 therapy is being contemplated for mast cell-mediated allergic disease.

2. I have several questions about internalization.

- i) It is described that a non-blocking antibody was used to detect amount remaining on the cell surface. Evidence needs to be presented for incubation between the clones and this non-blocking antibody with cells on ice that they don't interfere with each others binding.

- ii) It would have been more convincing if internalization of Siglec-6 is directly shown.

- iii) Are Fc/FcR interactions involved in internalization?

3. Figure 4D needs a lot more to convince me this is a real biological effect.

- i) Unstimulated cells need to be shown.

ii) Quantification over many images is required. "size of clusters induced by AK04 were significantly larger" cannot be concluded from the presented data alone.

4. Is co-engagement of activatory FcγR also involved in phosphorylation of Siglec-6?

Minor:

1. Siglec-6 does not have two ITIMs, but rather an ITIM and an ITIM-like sequence as they depict in Figure 1C. The text needs to be adjusted accordingly.

2. Figure 2A does not have error bars. Can we assume that one biological replicate and one technical replicate was carried out?

3. For the WBs, please put an arrow to indicate the band for the protein in question as there are multiple bands in some lanes.

RE: COMMSBIO-22-2116 Point-by-Point Response

REVIEWER 1:

Major Comments

1. In Figure 2A several antibodies are shown to induce complete endocytosis/internalization of Sig6 while others show effectively none (e.g. AK01 and AK04). Residual Sig6 remaining on the surface was detected using a commercially available ‘non-blocking’ anti-Sig6 antibody. The conclusions rely on the commercial antibody (AF647-labeled) to have no overlap in binding to Sig6 with the other antibodies. A simple control would be to do the experiment at 4° C where no internalization would be possible (e.g. 1 hr at 4°). If there is no competition, the AF647-anti-Sig6 binding should be the same for all antibodies. Otherwise, the data in Fig.2A could simply be a result of some antibodies preventing binding of the AF647-anti-Sig6.

Author response: We thank the Reviewer for bringing up this point that was also mentioned by Reviewer #3. We have provided more information and data on the Siglec-6 antibodies used to detect internalization, including data from the experiment the reviewers suggested (Supplemental Figure 3A and B). Siglec-6 mAb clones AK02 and AK05 were used to detect internalization. AK05 was used to detect internalization of all clones except AK05 and AK06, whereas AK02 was used to detect internalization of AK05 and AK06.

Lines 136-140: “To investigate if our panel of Siglec-6 mAbs mediated activity through Siglec-6, we first evaluated Siglec-6 mAb-induced receptor internalization by flow cytometry using peripheral blood-derived human MCs (hMCs). **Siglec-6 internalization was detected using two fluorophore-conjugated, non-competing Siglec-6 mAbs (AK05 and AK02) depending on the treatment mAbs (Supplemental Figure 3a, 3b).**”

2. The rationale for the internalization experiment with THP1 cells in Fig. 2C is not clear. AK04 was just demonstrated not to be an internalizing antibody in Fig. 2A. Now, it is said to internalize dependent on Fcγ receptors. Presumably this is meant to be Fcγ receptors on THP1 cells. Not clear how receptors on another cell can mediate internalization UNLESS it is meant that the THP1 cells phagocytose the mast cells. This is actually proposed in Fig. 6A. There is no clear explanation concerning how Fcγ receptors on THP1 cells can mediate AK04 internalization/endocytosis of Sig6.

Author response: We have clarified the rationale for using the THP-1 cells in the text and have provided additional data to support Fcγ receptors mediate AK04 internalization of Siglec-6. In addition, we have provided text in the discussion expanding on this mechanism.

Lines 148-149: “Fcγ receptor binding was recently shown to mediate programmed death-ligand 1 (PD-L1) receptor internalization for the PD-L1 mAb avelumab. To evaluate if Fcγ receptor binding was needed for Siglec-6 receptor internalization for the Bin A binders, we cultured hMCs alone or in the presence of the Fcγ receptor expressing THP-1 monocytic cell line with AK04 hIgG1 or an isotype control.”

Lines 175-180: “To confirm Fc receptors mediated AK04 internalization of Siglec-6, we co-cultured hMCs with Chinese hamster ovary (CHO) cells expressing the high affinity IgG receptor, CD64 (CHO-CD64) (Supplemental Figure 4a). AK04 induced dose-dependent Siglec-6 internalization in the presence of CHO-64 cells compared to an isotype control (Supplemental Figure 4b), demonstrating Fc receptor interaction is required for AK04 internalization.”

3. In Fig. 4, how does the expression of Sig6 in BMBCs compare to expression of human mast cells.

Author response: The expression of Siglec-6 on transfected BMBCs is 10x higher than hMCs. These data have been added as Supplemental Figure 6A.

Lines 230-231: “Siglec-6 expressing BMBCs displayed similar expression across constructs and higher levels than hMCs (Supplemental Figure 6a).”

A

4. Does AK04 reduction of anaphylaxis in Fig.5 relate to the induction phagocytosis by THP1 cells in Fig. 6? The mast cell counts in Fig. 5 are similar for AK04 and IsoType antibody, so it suggests not.

Author response: The reviewer is correct. The reduction in anaphylaxis does not seem to be due to phagocytosis as mast cell counts remain similar between groups. The decrease in anaphylaxis is most likely due to mast cell inhibition mediated through Siglec-6. The inhibitory vs phagocytosis activity of AK04 seems to be dependent on the number of doses with a single injection of mAb resulting in inhibition and repeat injections inducing lower mast cell numbers as is shown in Figure 6 E and F.

Minor Comments

1. No vendor is listed for CRA-1.

Author response: Thank you for pointing out, the vendor has been added.

REVIEWER 2:

Major Comments

1. The first results presented, regarding Siglec-6 expression on MCs from different cells/tissues, seem to be somehow redundant, as partially shown already in Robida et al., 2022. In addition, these results do not appear to be in line with the concept of the manuscript, which shows the characterization of these new anti-Siglec-6 antibodies. The first paragraph of the results should be included in the subsequent one, as proof of concept and reconfirmation that Siglec-6 is expressed specifically on MCs.

Author response: We thank the reviewer for the suggestion. The paragraphs have been combined and text modified.

2. In the MC activation assay, the Authors tested only one concentration of the anti-Siglec-6 antibody. It is possible that the lack of significant inhibition seen with some of the antibodies

might be due to not sufficient concentrations of the antibodies themselves. The Authors should discuss this point.

Author response: Great point. We have added additional data evaluating multiple concentrations of AK02 and AK04 on mast cell activation. These data are presented as Supplemental Figure 5C.

Lines 212-216: “Using a single concentration of CRA-1, we further evaluated the inhibitory activity of AK02 and AK04 on IgE-mediated MC activation. AK04 significantly reduced degranulation of FcεRI-mediated MC activation compared to AK02 (Figure 3c). Similar findings were seen through titration studies with both Siglec-6 mAb clones, confirming AK04 mediates more potent MC inhibition (Supplemental Figure 5c).”

C

3. It is not clear why the Authors chose to employ the AK02 antibody, considering that AK01 showed better inhibition of MC activation. This point has to be discussed.

Author response: We have clarified this better in the text. The rationale is provided below:

Lines 210-212: “To better understand Siglec-6 epitope-dependent agonism, we focused AK02 and AK04 because these clones bind to domain 1 with similar affinities, but display different internalization properties, with AK04 requiring Fc-interaction.”

4. Did the Authors check the release of soluble mediators from MCs? This would be important to understand whether engagement of Siglec-6 by mAbs influences MC degranulation.

Author response: We agree and yes, we measured tryptase and IL-8 production from MCs treated with AK02 and AK04. We’ve made this clearer in the text and the data is shown in Figure 3 panel D.

Lines 216-218: “In addition to reduced degranulation, AK04 significantly decreased soluble mediator production of FcεRI-activated hMCs, including tryptase and IL-8 compared to AK02 (Figure 3d).

D

5. How do the Authors explain MC ADCP, when co-culture of MCs and THP-1 cells resulted in Siglec-6 internalization? How is phagocytosis of MCs triggered, if the antibody is internalized with the receptor? The Authors should discuss this point.

Author response: This is an interesting question. The internalization of Siglec-6 occurs over several hours and the receptor is recycled to the surface in a similar time frame. Based on this, we believe there is sufficient time for the mAb to engage with effector cells. We have expanded on this point in the discussion.

Lines 415-418: “It is interesting to note that AK04 mediates ADCP of MCs while also inducing Siglec-6 internalization, suggesting the kinetics of Siglec-6 internalization and recycling are sufficient for Fc-interaction with effector cells. In support of this, lirentelimab, a Siglec-8 mAb that causes rapid receptor internalization is still capable of ADCC activity (46).

6. Did the Authors evaluate MC repopulation in the tissues after depletion with the antibody?

Author response: This is a great idea. We did not evaluate mast cell repopulation in our models but will consider monitoring in future experiments.

7. The Authors detected Siglec-6 only on MCs but not on other immune cells in the tissues, and on trophoblasts, which was previously reported as a human-specific occurrence (Brinkman-Van der Linden et al., 2007). The Authors should discuss more the fact that Siglec-6 expression in humans is so specific.

Author response: We have incorporated additional data on the selectivity of Siglec-6 and expanded on this in the discussion.

Lines 355-360: “Our findings corroborate previous studies showing Siglec-6 is highly and selectively expressed on tissue MCs and to a much lower extent, memory B cells in non-malignant samples (21,27,28,33). Despite detectable levels of Siglec-6 on B cells, Siglec-6 mAb treatment did not induce ADCP of memory B cells. These findings are consistent with

the ‘threshold’ phenomenon, whereby high levels of surface antigen are required for optimal antibody-mediated effector mechanisms, such as ADCP and ADCC (34).”

REVIEWER 3:

Major Comments

1. The authors have missed that Siglec-6 is also expressed on memory B cells. Depleting memory B cells would completely erase immunological memory and be a major limitation of anti-Sig6 dependent depletion of mast cells. At minimum, 'off-target' effects on memory B cells need to be assessed if an anti-Sig6 therapy is being contemplated for mast cell-mediated allergic disease.

Author response: We thank the reviewer for bringing this reference to our attention. We have provided additional data evaluating the expression of Siglec-6 on human peripheral blood immune cells, including unswitched and switched memory B cells. Siglec-6 was found to be expressed at low levels on memory B cells compared to tissue mast cells (Supplemental Figure 1c). To address the potential depletion of memory B cells with a Siglec-6 mAb, we performed ADCP assays with isolated human B cells and used Rituximab as a positive control (Supplemental Figure 7). In addition, we have added text in the discussion regarding this point.

Lines 59-67: “Siglec-6 has been reported to be expressed on skin and esophageal tissue MCs, specific populations of trophoblasts, and memory B cells (21,27,28). To confirm Siglec-6 expression, we profiled Siglec-6 surface expression on major immune cell populations in human peripheral blood as well as lung, skin, and gastrointestinal (GI) tissues by flow cytometry using a commercially available mAb (Supplemental Figure 1a and 1b). Siglec-6 expression was consistently detected at high levels on MCs from all tissues evaluated (~6000 dMFI) (Supplemental Figure 1c). In addition to MCs, low levels of Siglec-6 were found on unswitched (~250 dMFI) and switched (~450 dMFI) memory B cells (Supplemental Figure 1c). Siglec-6 expression was not found on any other immune cells in blood or tissues”

Lines 329-332: “Since memory B cells had low, but detectable expression of Siglec-6, we next evaluated AK04-mediated ADCP activity of human primary B cells using THP-1 cells. As a positive control, we titrated rituximab, an anti-CD20 mAb with known ADCP activity against B cells. Rituximab, but not AK04, induced ADCP of CD19⁺ B cells and reduced CD27⁺ memory B cell counts compared to an isotype control (Supplemental Figure 7).”

Lines 355-360: “Our findings corroborate previous studies showing Siglec-6 is highly and selectively expressed on tissue MCs and to a much lower extent, memory B cells in non-malignant samples (21,27,28,33). Despite detectable levels of Siglec-6 on B cells, Siglec-6 mAb treatment did not induce ADCP of memory B cells. These findings are consistent with the ‘threshold’ phenomenon, whereby high levels of surface antigen are required for optimal antibody-mediated effector mechanisms, such as ADCP and ADCC (34).”

2. It is described that a non-blocking antibody was used to detect amount remaining on the cell surface. Evidence needs to be presented for incubation between the clones and this non-blocking antibody with cells on ice that they don't interfere with each other's binding.

Author response: Reviewer #1 also brought up the same point. We kindly direct the reviewer to our response for question 1 from Reviewer #1.

3. It would have been more convincing if internalization of Siglec-6 is directly shown.

Author response: Great point. We have added live confocal videos of Siglec-6 internalization on hMCs treated with either AK02 or AK04 (Supplemental Video 1).

4. Are Fc/FcR interactions involved in internalization?

Author response: Yes, Fc/FcR interaction is required for AK04-mediated internalization of Siglec-6. We have provided additional data to support this mechanism.

Lines 175-180: “To confirm Fc receptors mediated AK04 internalization of Siglec-6, we co-cultured hMCs with Chinese hamster ovary (CHO) cells expressing the high affinity IgG receptor, CD64 (CHO-CD64) (Supplemental Figure 4A). AK04 induced dose-dependent Siglec-6 internalization in the presence of CHO-64 cells compared to an isotype control (Supplemental Figure 4B), demonstrating Fc receptor interaction is required for AK04 internalization.”

5. Figure 4D needs a lot more to convince me this is a real biological effect - Unstimulated cells need to be shown.

Author response: Great suggestion. We have included untreated cells in Supplemental Figure 6B and modified the text

Lines 270-284: “BMMCs were transfected with WT or double ITIM mutant Siglec-6 expression plasmids, treated with a Siglec-6 mAb and subjected to confocal microscopy. In the untreated state, minimal co-localization of Shp-1 and Siglec-6 was seen in either WT or double ITIM mutant transfected BMMCs (Supplemental Figure 6b). Strikingly, Siglec-6

mAb-treatment resulted in Shp-1 co-localization in WT expressing Siglec-6 clusters, but not in the double ITIM mutant Siglec-6 clusters (Figure 4d).”

6. Figure 4D - Quantification over many images is required. “size of clusters induced by AK04 were significantly larger” cannot be concluded from the presented data alone.

Author response: This comment specifies Figure 4D which examines the interaction of Shp1 and Siglec-6 in BMDCs transfected with WT and mutant ITIM Siglec-6. We do not mention Siglec-6 cluster size in this figure.

The experiment in Figure 3E-F quantifies cluster size of Siglec-6 specific clones. To that end, we have used fluorescently conjugated AK02 and AK04. Once added to the media, we immediately begin imaging, thus time point zero is the closest we can get to visualizing an unstimulated state in this experiment without performing fixation. In both cases, AK02 and AK04 surface staining look identical before clustering. The figures show a zoom of the imaging field for representative image purposes. However, in total we have analyzed 120 cells for each condition across a time of 4 hours. We believe this is sufficient to conclude that cluster sizes are larger for AK04. To help facilitate this, we have added a supplementary video (Sup Movie 1) and provided stats for Figure 3F.

7. Is co-engagement of activatory FcγR also involved in phosphorylation of Siglec-6?

Author response: This is an interesting question that we are currently investigating. We have not yet been able to show direct phosphorylation of Siglec-6 ITIM/ITIM-like motifs upon FcγR-mediated crosslinking, but we know that for effective inhibition through antibody mediated cross-linking (using a secondary antibody) the phosphorylation of these tyrosine-based motifs is required (Figure 4) and it is possible that engagement through activating FcγR could facilitate this also.

Minor Comments

1. Siglec-6 does not have two ITIMs, but rather an ITIM and an ITIM-like sequence as they depict in Figure 1C. The text needs to be adjusted accordingly.

Author response: Thank you for pointing this out. We have corrected the text.

2. Figure 2A does not have error bars. Can we assume that one biological replicate and one technical replicate was carried out?

Author response: We have updated the figure legend to reflect the number of replicates performed.

Figure 2A legend: (a) Siglec-6 internalization on hMCs after overnight incubation with the indicated Siglec-6 mAb clones as determined by flow cytometry using AK05 as the detection mAb. Data are representative of 2 experiments with 2 independent donors.

3. For the WBs, please put an arrow to indicate the band for the protein in question as there are multiple bands in some lanes.

Author response: Great idea, this figure has been updated with arrows.

Reviewers' comments:

Reviewer #1 (Remarks to the Author):

The clarifications and additional experiments largely satisfy the major concerns of this referee, and provide needed support for the conclusions. One major point is still unclear. If antibody bound to Siglec-6 on a mast cell encounters FcγR on THP-1 cells, intuitively it should stabilize the anti-Sig6/Sig6 complex at the interface of the two cells (or cause phagocytosis). Instead it induces endocytosis of the complex by the mast cell? At a minimum, a clear rationale for why this happens should be offered.

Have the authors considered the alternative that the Fc receptor on THP-1 cells or CHO cells extracts the anti-Sig6/Sig6 from the membrane of the mast cell?

Reviewer #2 (Remarks to the Author):

The Authors have addressed all the points raised by this Reviewer. No further amendments are required. The manuscript is suitable for publication.

Reviewer #3 (Remarks to the Author):

The authors have largely satisfied my previous comments. My only concern is the Sig6 expression levels in Supp Fig 1c. It would be much more thorough to compare to an isotype control. Moreover, I cannot tell if an Fc block was conducted. This is important given that the authors have demonstrated that these anti-Sig6 antibodies are Fc engagers. It would not be necessary to repeat all the staining, but perhaps just the mast cell staining to ensure the Sig6 expression levels are not significantly impacted by not including these controls.

RE: COMMSBIO-22-2116A Point-by-Point Response

REVIEWER 1:

Major Comments

1. The clarifications and additional experiments largely satisfy the major concerns of this referee, and provide needed support for the conclusions. One major point is still unclear. If antibody bound to Siglec-6 on a mast cell encounters Fc γ R on THP-1 cells, intuitively it should stabilize the anti-Sig6/Sig6 complex at the interface of the two cells (or cause phagocytosis). Instead it induces endocytosis of the complex by the mast cell? At a minimum, a clear rationale for why this happens should be offered.

Have the authors considered the alternative that the Fc receptor on THP-1 cells or CHO cells extracts the anti-Sig6/Sig6 from the membrane of the mast cell?

Author response: We thank the Reviewer for these insightful questions. We have provided a rationale in the discussion as well as additional data to support our hypothesis. We hypothesize that the interaction between AK04 and THP-1 cells is stabilized for a short period of time in which the THP-1 cells either phagocytose mast cells or induce internalization. In support of this, we have added data evaluating the kinetics of AK04-induced internalization where peak internalization is seen around 4 hours post antibody treatment. Since ADCP has been reported to occur quickly, we think there is sufficient time for THP-1 cells to induce ADCP. In addition, Siglec-6 is not fully internalized with AK04, which could present additional opportunities for ADCP of mast cells over time. Regarding other mechanisms of Fc-mediated macrophage activity, such as trogocytosis, we have performed preliminary experiments and not seen evidence of this. We've also mentioned this in the discussion.

Lines 122-125: “To understand the kinetics of AK04-mediated Siglec-6 internalization, hMCs were cultured in the presence of THP-1 cells and internalization was monitored over 24 hours. Siglec-6 internalization occurred 1-hour post-AK04 treatment and peaked around 4 hours (Supplemental Figure 3e).”

Lines 310-318: “We hypothesize that the interaction between the Fc-region of AK04 and THP-1 cells is stabilized for a short period of time in which the THP-1 cells either phagocytose MCs or induce internalization. In support of this, AK04-induced internalization peaks around 4 hours which should provide sufficient time for ADCP as this process has been

reported to occur rapidly. In addition, Siglec-6 is not completely internalized on the MC surface, providing additional opportunity for ADCP over time. However, additional studies are needed to better understand the kinetics of AK04-induced internalization and ADCP of MCs. While our findings support AK04 induces ADCP of mast cells via macrophages, we have not ruled out other mechanisms of macrophage activity, such as trogocytosis.”

REVIEWER 3:

Major Comments

1. The authors have largely satisfied my previous comments. My only concern is the Sig6 expression levels in Supp Fig 1c. It would be much more thorough to compare to an isotype control. Moreover, I cannot tell if an Fc block was conducted. This is important given that the authors have demonstrated these anti-Sig6 antibodies are Fc engagers. It would not be necessary to repeat all the staining, but perhaps just the mast cell staining to ensure the Sig6 expression levels are not significantly impacted by not including these controls.

Author response: We thank the Reviewer for their thorough evaluation of the manuscript. Fc block was used in the immunophenotyping experiments to prevent non-specific binding. This is stated in the methods (lines 447-448), and we have added more text to expand on this. In addition, as suggested, we have evaluated Siglec-6 expression on mast cells in 4 different human lung tissue donors using both an isotype control PE fluorophore conjugated antibody and FMO. The raw MFI and delta MFI (dMFI) data are shown below. Similar raw and normalized expression levels of Siglec-6 are seen using either FMO or isotype control on human tissue mast cells.

Siglec-6 Expression on Human Tissue Lung Mast Cells

Figure: Human lung tissue (n=4 donors) was processed into single cells and stained to identify mast cells as previously described by flow cytometry. (A) Cells were either stained with an FMO (black), isotype control PE IgG2a antibody (gray, Biolegend), or Siglec-6 PE IgG2a antibody (orange, R&D Systems) and analyzed by flow cytometry to evaluate expression (MFI). (B) Siglec-6 expression as shown as delta MFI whereby raw MFI values are subtracted from the FMO control.

REVIEWERS' COMMENTS:

Reviewer #1 (Remarks to the Author):

While it is still not clear how THP-1 cells impact internalization of AK04, the additional discussion of alternatives satisfies this reviewers concerns.

Reviewer #3 (Remarks to the Author):

Authors have adequately responded to reviewers comments.